# Fungal Species from *Rhododendron* sp.: *Discosia rhododendricola* sp.nov, *Neopestalotiopsis rhododendricola* sp.nov and *Diaporthe nobilis* as a New Host Record.

**DOI:** 10.3390/jof8090907

**Published:** 2022-08-26

**Authors:** Napalai Chaiwan, Rajesh Jeewon, Dhandevi Pem, Ruvishika Shehali Jayawardena, Nadeem Nazurally, Ausana Mapook, Itthayakorn Promputtha, Kevin D. Hyde

**Affiliations:** 1Center of Excellence in Fungal Research, Mae Fah Luang University, Chiang Rai 57100, Thailand; 2CAS Key Laboratory for Plant Diversity and Biogeography of East Asia, Kunming Institute of Botany, Chinese Academy of Sciences, Kunming 650201, China; 3Department of Health Sciences, Faculty of Medicine and Health Sciences, University of Mauritius, Reduit 80837, Mauritius; 4Department of Agricultural and Food Science, Faculty of Agriculture, University of Mauritius, Reduit 80837, Mauritius; 5Department of Biology, Faculty of Science, Chiang Mai University, Chiang Mai 50200, Thailand; 6Environmental Science Research Center, Faculty of Science, Chiang Mai University, Chiang Mai 50200, Thailand; 7Innovative Institute of Plant Health, Zhongkai University of Agriculture and Engineering, Haizhu District, Guangzhou 510225, China

**Keywords:** leaf litter, multi-loci phylogenetic analyses, new taxa, saprobe, *Sordariomycetes*, taxonomy

## Abstract

In the present study, we report two new asexual fungal species (i.e., *Discosia rhododendricola*, *Neopestalotiopsis rhododendricola* (*Sporocadaceae*) and a new host for a previously described species (i.e., *Diaporthe nobilis*; *Diaporthaceae*). All species were isolated from *Rhododendron* spp. in Kunming, Yunnan Province, China. All taxa are described based on morphology, and phylogenetic relationships were inferred using a multigenic approach (LSU, ITS, *RPB2*, *TEF1* and *TUB2*). The phylogenetic analyses indicated that *D. rhododendronicola* sp. nov. is phylogenetically related to *D. muscicola*, and *N. rhododendricola* sp. nov is related to *N. sonnaratae*. *Diaporthe nobilis* is reported herein as a new host record from *Rhododendron* sp. for China, and its phylogeny is depicted based on ITS, *TEF1* and *TUB2* sequence data.

## 1. Introduction

*Rhododendron*, a genus of shrub and small to large trees belonging to *Ericaceae*, is an indicator of health for forest areas [1], commonly found in low-quality acidic soil and sterile conditions. This plant is mainly distributed in India and southeastern Asia, extending from the northwest Himalayas (Arunachal Pradesh) to Bhutan, eastern Tibet, Nepal, north Myanmar, Sikkim, and west central China [2]. *Rhododendron* flowers are used as food, to produce fermented wine, and to make herbal tea due to their distinctive flavor and color [3,4]. Fungi colonizing *Rhododendron* include *Alternaria alternata*, *Aspergillus brasiliensis*, *Chrysomyxa dietelii*, *C. succinea* [5], *Diaporthe nobilis* [6], *Epicoccum nigrum*, *Mucor hiemalis*, *Pestalotiopsis sydowiana* and *Trichoderma koningii* [7]. However, given the economic importance of this plant, it is imperative to assess the fungal species associated with it.

*Discosia* was introduced in Discosiaceae by Maharachchikumbura et al. [8] to accommodate the type genus *Discosia* and the type species *D. artocreas.* Senanayake et al. [9,10] introduced *Adisciso*, *Discosia*, *Discostroma*, *Immersidiscosia*, *Sarcostroma* and *Seimatosporium* in the *Discosiaceae* family. Jaklitsch et al. [11] considered *Discosiaceae* a synonym of *Sporocadaceae* based on DNA sequence analyses with strong phylogenetic support. Wijayawardene et al. [12] accepted *Discosia* species belonging to the family *Sporocadaceae*. Libert introduced *Discosia* in 1837, with *Discosia strobilina* being the lectotype [9,12]. Liu et al. [13] reviewed the generic description of *Discosia*, an updated morphology, and the phylogenetic relationships based on ITS sequence data [13]. There are 118 epithets of *Discosia* in Index Fungorum 2022 [14]. *Discosia* has been identified as an asexual fungus and is characterized by uni- to multilocular conidiomata with muti-layered walls. Conidiogenous cells are monoblastic and phialidic to annellidic. Conidial types are bipolar, polar and subpolar appendages, and usually hyaline to pale brown [9].

The genus *Neopestalotiopsis*, introduced by Maharachchikumbura et al., (2014) [15], belongs to the family Sporocadaceae (Amphisphaeriales, Sordariomycetes) [8,15,16], with *N. protearum* being the type species. *Neopestalotiopsis* species have been reported on saprobes, trees or plant pathogens causing postharvest diseases (fruit rots and leaf blights) [17,18]. The sexual morph of *Neopestalotiopsis* species remains unknown [15,18,19,20]. *Neopestalotiopsis* species have a worldwide distribution. This genus has also been reported in caves in China [15,17,18,19,20,21,22]. Studies related to the taxonomy of *Neopestalotiopsis* included DNA sequence analyses and phylogeny of the ITS, *TEF1* and *TUB2* [22].

The genus *Diaporthe* introduced by Nitschke [23] belongs to the families Diaporthaceae, Diaporthales, and Sordariomycetes [8,9]. *Diaporthe* species have a worldwide distribution [6,24,25,26,27,28]. This genus has been associated with several grapevine diseases in Europe [29] and was detected in Uruguayan deciduous fruit tree (Malus domestica ‘Gala’) wood disease [30]. Studies related to the taxonomy of *Diaporthe* included DNA sequence analyses and phylogeny of the ITS, *TEF1*, *TUB2* and *CAL* loci [6,31]. Dissanayake et al. [32] provided phylogenetic relationships of 171 *Diaporthe* species currently known from culture or direct sequencing, and are linked to their holotype, epitype, isotype or neotype and that can now be recognized with DNA sequence data, essential to species identification [33].

In this study, we introduce the new species *D. rhododendricola*, *N. rhododendricola* and a new host record of *Diaporthe nobilis*, collected from dead leaves of *Rhododendron* species in China. We further provide descriptions, illustrations, and DNA sequence-based phylogeny to verify identification and placement.

## 2. Materials and Methods

### 2.1. Sample Collection, Morphological Observation, and Fungal Isolation

Isolation was performed as described by Senanayake et al. [34]. Dead leaves of *Rhododendron* spp. were collected from Kunming, Yunnan Province, China and brought to the laboratory in labelled paper envelopes. A light microscope (Nikon ECLIPSE 80i compound microscope, Melville, NY, USA) was used to observe the specimens. Spore mass fruiting bodies were isolated on potato dextrose agar (PDA) plates and incubated at 25 °C.

The isolates were transferred to new PDA plates, incubated at 25 °C, and photographed using a Canon EOS 600D digital camera fitted to the microscope. The Tarosoft (R) Image Frame Work program measured the morphological characteristics. The figures were processed using Adobe Photoshop CS6 Extended version 10.0 (Adobe Systems, San Jose, CA, USA).

The specimens were deposited at the Herbarium of Mae Fah Luang University (Herb. MFLU) and Herbarium of Kungming Institute of Botany (KUN), Chinese Academy of Science, Kunming, China. Living cultures were deposited at the Culture Collection of Mae Fah Luang University (MFLUCC), Chiang Rai, Thailand and the Culture Collection of Kungming Institute of Botany (KUN), Chinese Academy of Science, Kunming, China. Faces of Fungi and Index Fungorum data are also provided [14,35]. New species were established based on guidelines provided by Jeewon and Hyde [36].

### 2.2. DNA Extraction, PCR Amplification, and Sequencing

Fungal cultures were grown on PDA at 25 °C for 2–4 weeks. The Biospin Fungus Genomic DNA Extraction Kit-BioFlux (BioFlux^®^, Hangzhou, China) was used to extract DNA from the mycelium. PCR amplification was performed using primer pairs, ITS4/ITS5 for the internal transcribed spacer region of ribosomal DNA [37], LR0R/LR5 for large subunit nuclear ribosomal DNA [38], EF-728F/EF-986R for translation elongation factor 1-alpha gene [39], fRPB2-5f/fRPB2-7cR for the second largest subunit of RNA polymerase [40] and Bt2a/Bt2b for beta-tubulin [41]. The PCR conditions were based on the methodology as described by Chaiwan et al. [42].

### 2.3. Phylogenetic Analyses

The sequence alignment and phylogenetic analyses were performed as outlined by Dissanayake et al. [43] and Chaiwan et al. [42,44,45]. Phylogenetic analyses were performed using a combined *Discosia* dataset of ITS, LSU, *RPB2*, *TEF1* and *TUB2* sequence data and a combined *Neopestalotiopsis* and *Diaporthe* dataset of ITS, *TEF1* and *TUB2* sequence data. Taxa used in the analyses were obtained through recent publications [16,28,46]. The phylogenetic analyses were carried out using maximum parsimony (MP), maximum likelihood (ML) and Bayesian posterior probabilities (BYPP). PAUP v4.0b10 was used to conduct the parsimony analysis to obtain the phylogenetic trees [47]. Trees were inferred using the heuristic search option with 1000 random sequence additions. Maxtrees were set to 1000, branches of zero length were collapsed and all multiple parsimonious trees were saved. Descriptive tree statistics for parsimony—tree length (TL), consistency index (CI), retention index (RI), relative consistency index (RC) and homoplasy index (HI)—were calculated for trees generated following the Kishino-Hasegawa test (KHT) criteria [48], which was performed in order to determine whether trees were significantly different. Maximum-parsimony bootstrap values equal or greater than 60% are given as the second set of numbers above the nodes.

Maximum likelihood analysis was performed by using RAxML-HPC2, New Orleans, LA on XSEDE (8.2.8) [45,48,49,50]. The search strategy was set to rapid bootstrapping and the analysis was carried out using the GTRGAMMAI model of nucleotide substitution. Maximum likelihood bootstrap values equal to or greater than 60% are given as the first set of numbers above the nodes.

Bayesian inference (BI) analysis was conducted with MrBayes v. 3.1.2 to evaluate the posterior probabilities (BYPP) using Markov chain Monte Carlo sampling [51]. Two parallel runs were conducted using the default settings, but with the following adjustments: six simultaneous Markov chains were run for 2,000,000 generations and trees were sampled every 200 generations. The distribution of log-likelihood scores were examined to determine stationary phase for each search and to decide if extra runs were required to achieve convergence, using the program Tracer 1.4 [52]. The first 10% of generated trees were discarded and the remaining 90% of trees were used to calculate posterior probabilities (PP) of the majority rule consensus tree. The phylogenetic trees were viewed in FigTree v. 1.4 [53] and edited using Microsoft Office Power Point 2007 and Adobe Photoshop CS6 Extended [42].

### 2.4. Genealogical Concordance Phylogenetic Species Recognition (GCPSR) Analysis

The related species were analyzed using the Genealogical Concordance Phylogenetic Species Recognition model. The pairwise homoplasy index (PHI) [54] is a model test based on the fact that multiple gene phylogenies will be concordant between species and discordant due to recombination and mutations within a species. The data were analyzed by the pairwise homoplasy index (PHI) test [54]. The test was performed in SplitsTree4 [55,56] as described by Quaedvlieg [57] to determine the recombination level within phylogenetically closely related species using a five-locus concatenated dataset to determine the recombination level within phylogenetically closely related species. If the PHI is below the 0.05 threshold (Φw < 0.05), it indicates that there is significant recombination in the dataset. This means that related species in a group and recombination level are not different. If the PHI is above the 0.05 threshold (Φw > 0.05), it indicates that it is not significant, which means the related species in a group level are different. The new species and its closely related species were analyzed using this model. The relationships between closely related species were visualized by constructing a split graph, using both the LogDet transformation and splits decomposition options.

### 2.5. Discosia, Habitat and Known Distribution Checklist Associated with Rhododendron sp.

An updated checklist of *Discosia* based on the SMML database (https://nt.ars-grin.gov/fungaldatabases/) (accessed on 10 June 2022) is provided [58]. Those species for which molecular data are available are indicated. The distribution information regarding the type or original descriptions available and the locality from which *Discosia* have been recorded on *Rhododendron* spp. is provided, including all the specimens encountered during this study.

## 3. Results

### 3.1. Phylogenetic Analyses

The combined sequence alignments of *Discosia* comprised 54 taxa (Table 1), with *Immersidiscosia eucalypti* MFLU16-1372 and NBRC 104195 as the outgroup taxa. The dataset comprised 4364 characters including alignment gaps (LSU, ITS, *RPB2*, *TEF1* and *TUB2* sequence data). The MP analysis for the combined dataset had 430 parsimony-informative, 3522 constant, and 412 parsimony-uninformative characters, and yielded a single most parsimonious tree (TL = 1353, CI = 0.777, RI = 0.764, RC = 0.594; HI = 0.223). The RAxML analysis of the combined dataset yielded a best scoring tree with a final ML optimization likelihood value of −22,013.917605. The matrix had 840 distinct alignment patterns, with 66% undetermined characters or gaps. Bayesian posterior probabilities from Bayesian inference analysis were assessed with a final average standard deviation of split frequencies = 0.009983. The phylogenetic tree in this study showed that our strain (*Discosia rhododendricola* KUN-HKAS 123205 and MFLU20-0486) is related to *D. muscicola* with high support value in the phylogenbetic tree (Figure 1). Sequence alignments are deposited in TreeBASE.

The combined sequence alignments of *Neopestalotiopsis* comprised 89 taxa (Table 2), with *Monochaetia monochaeta* CBS115004 and *M. ilexae* CBS101009 as the outgroup taxa. The dataset comprised 2634 characters including alignment gaps (ITS, *TUB2* and *TEF1* sequence data). The MP analysis for the combined dataset had 631 parsimony-informative, 1524 constant, and 479 parsimony-uninformative characters, and yielded a single most parsimonious tree (TL = 2304, CI = 0.679, RI = 0.813, RC = 0.552; HI = 0.321). The RAxML analysis of the combined dataset yielded a best scoring tree with a final ML optimization likelihood value of −24,500.881631. The matrix had 1268 distinct alignment patterns, with 35.77% undetermined characters or gaps. Bayesian posterior probabilities from Bayesian inference analysis were assessed with a standard deviation of split frequencies = 0.024223. The phylogenetic tree in this study showed that *N. rhododendricola* KUN-HKAS 123204 and MFLU20-0046 belonged to a separate clade, phylogenetically related to *N. sonneratae*, *N. coffeae-arabicae* and *N. thailandica* with 88% MP support (Figure 2). Sequence alignments are deposited in TreeBASE.

The combined sequence alignments of *Diaporthe* comprised 56 taxa (Table 3), with *Diaporthella corylina* CBS 121124 used as the outgroup taxon. The dataset comprised 2350 characters, including alignment gaps (ITS, *TEF1* and *TUB2* sequence data). After alignment, 641 characters were derived from ITS, 916 from *TEF1*, and 793 from *TUB2*. The MP analysis for the combined dataset had 730 parsimony-informative, 1216 constant, and 404 parsimony-uninformative characters, and yielded a single most parsimonious tree (TL = 3968, CI = 0.480, RI = 0.622, RC = 0.298; HI = 0.520). The RAxML analysis of the combined dataset yielded a best scoring tree with a final ML optimization likelihood value of −21,299.667012. The matrix had 1319 distinct alignment patterns, with 37.51% undetermined characters or gaps. Estimated base frequencies were as follows: A = 0.226983, C = 0.316389, G = 0.231894, T = 0.224734; substitution rates AC = 1.153998, AG = 3.111864, AT = 1.039115, CG = 0.869376, CT = 4.271324, GT = 1.000000; gamma distribution shape parameter α = 0.376625. Bayesian posterior probabilities from Bayesian inference analysis were assessed with a standard deviation of split frequencies = 0.009867. The phylogenetic tree in this study showed that *D. nobilis* KUN-HKAS 123203 grouped with the ex-type strain of *D. nobilis*, and formed a supported clade with 0.99 PP (Figure 3). Sequence alignments are deposited in TreeBASE.

### 3.2. Taxonomy

#### 3.2.1. *Discosia rhododendricola* Chaiwan & K.D. Hyde, sp. Nov. (Figure 4)

MycoBank number: 845145; Facesoffungi number: FoF 09452

Etymology: name reflects the host from which the fungus was isolated.

Holotype: KUN-HKAS 123205

**Figure 4 jof-08-00907-f004:**
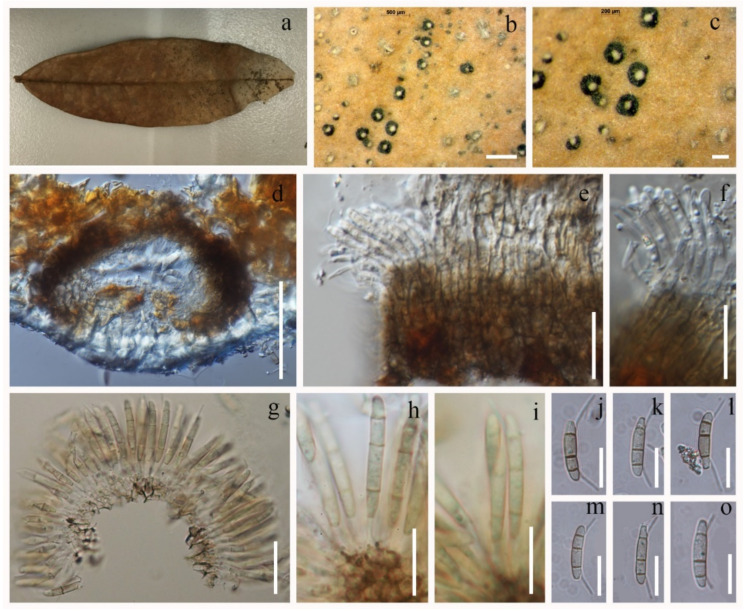
*Discosia rhododendricola* (KUN-HKAS 123205, holotype). (**a**–**c**) Appearance on host surface; (**d**) vertical section of conidioma; (**e**–**i**) conidiogenous cells and developing conidia; (**j**–**o**) conidia from holotype. Scale bars: (**b**) 500 μm; (**c**) 200 μm; (**d**) 100 μm; (**g**–**i**) 50 μm; (**e**,**j**–**o**) 20 μm; (**f**) 10 μm.

Saprobic on dead leaves of *Rhododendron* sp. **Sexual morph**: Undetermined. **Asexual morph**: *Conidiomata* 200–250 × 30–75 μm, pycnidial, cervular, applanate to disc-like, partly immersed or superficial, black, rounded to irregular in outline, glabrous, unilocular or divided into several locules by tissue conspicuous at the surface. *Conidiophores* were observed arising from the base, hyaline, filiform to cylindrical, smooth, and reduced to conidiogenous cells. *Conidiogenous* cells appeared subcylindrical, flask-shaped, hyaline, smooth, phialidic, each producing a single unbranched conidium. *Conidia* 20–30 × 4–5 μm ($\bar{x}$ = 25 × 4.5 μm, n = 30), subcylindrical, slightly curved, 3-septate, with slight constrictions at the septa, brown, smooth-walled with unequal cells; bipolar appendages; with a long, tubular base, two median cells subcylindrical, second cell joined to the base, 10–15 μm ($\bar{x}$ = 12.5 μm) long, the third cell joined to the apex, 11–15 μm ($\bar{x}$ = 13 μm) long; apical cell subconical with a rounded apex; apical and basal cells each with a subapical, unbranched, filiform, straight appendage; apical appendage, 9–11 μm ($\bar{x}$ = 10 μm), basal appendage, 20–25 μm ($\bar{x}$ = 22.5 μm).

Culture characteristics: Colonies grown on PDA were filamentous, raised, filiform margin, reached 4–5 cm in 5 days at 25 °C, brown to black, mycelium superficial, branched, septate, white mycelium with aerial on the surface, and produced black spore mass.

Material examined: CHINA, Kunming Yunnan Province; on dead leaves of *Rhododendron* sp. (Ericaceae), 28 July 2018, Napalai Chaiwan, KIB009 (KUN-HKAS 123205, holotype; isolate MFLU20-0486; Ex-type living culture KUNCC22-10804, isolate MFLUCC21-0004.

Notes: *Discosia rhododendricola* is similar to *D. macrozamiae* CPC 32109 [94] with regards to conidiomata size (*D. rhododendricola* a: 200–250 μm diam., 30–75 μm high vs. *D. macrozamiae* CPC 32109: 250 μm diam, 50 μm height). *Discosia rhododendricola* and *D. artocreas* (type species) share similar conidiophores lining the inner cavity (0–2-septate, rarely branched at base). There are also similar in conidial characteristics (conidial dimensions between 30 and 32 μm; the second cell joining to the base was 10–15 μm in length ($\bar{x}$ = 12.5 μm) in *D. rhododendricola*; 10–11 μm ($\bar{x}$ = 10.5 μm) in *D. macrozamiae* CPC 32109; the third cell joining to the apex was 11–15 μm in length ($\bar{x}$ = 13 μm) in *D. rhododendria* and 4–5 μm ($\bar{x}$ = 4.5 μm) in *D. macrozamiae* CPC 32109. The apical appendage of *D. rhododendricola* was 9–11 μm in length ($\bar{x}$ = 10 μm), while in *D. macrozamiae* (CPC 32109) it was 7–11 μm ($\bar{x}$ = 9 μm). The basal appendage in *D. rhododendricola* was 20–25 μm ($\bar{x}$ = 22.5 μm) in length, and in *D. macrozamiae* (CPC 32109) 10–16 μm ($\bar{x}$ = 13 μm).

*Discosia rhododendricola* differs from the type species, *D. artocreas*, in ascomatal size (*D. rhododendricola*: 200–250 μm diam., 30–75 μm high; *D. artocreas* 150–500 μm diam, 60 μm high). The two species share similar conidiophores and conidiogenous cells characteristics. However, *D. rhododendricola* has hyaline to pale brown conidiogenous cells and conidia, whereas *D. artocreas* has hyaline conidiogenous cells and conidia. The second cell joining to the base measured 10–15 μm in length ($\bar{x}$ = 12.5 μm) in *D. rhododendricola* but 5–9 μm ($\bar{x}$ = 7.5 μm) in *D. artocreas*. The third cell joining to the apex was 11–15 μm in length ($\bar{x}$ = 13 μm) in *D. rhododendria* and 3–6 μm ($\bar{x}$ = 4.5 μm) in *D. artocreas*. The apical appendage of *D. rhododendricola* was 9–11 μm in length ($\bar{x}$ = 10 μm), while in *D. artocreas* it was 6–12 μm ($\bar{x}$ = 10 μm). The basal appendage in *D. rhododendricola* was 20–25 μm in length ($\bar{x}$ = 22.5 μm), while in *D. artocreas* it was 7–12 μm ($\bar{x}$ = 10 μm).

The NCBI BLAST search of ITS sequence *D. rhododendricola* presented 95.32% similarity with *Immersdiscosia eucalypti*. A comparison of the 542 ITS (+5.8S) nucleotides of *D.rhododendricola* sp. nov. and *I. eucalypti* reveals 21 (3.87%) nucleotides differencess. We compared 876 LSU nucleotides of *D. rhododendricola* with *D. muscicola* CBS 109.48, and a 0.34% bp difference was observed (a difference of 3 bp in a total 879 bp) (Table 4).

When analyzing the sequences, *D. rhododendricola* sp. nov. (KUN-HKAS123205 and MFLU20-0486) were found to be phylogenetically related to *D. macrozamiae* CPC 32109, *D. muscicola* CBS 109.48, *D. pleurochaeta* KT2179, *D. pleurochae* KT 2188 and KT 2192, while *D. tricellularis* MAFF237478 and NCBR32705 and *D. yakushimensis* MAFF 242774 were found to be in a clade. The two isolates of the new taxon (KUN-HKAS123205 and MFLU20-0486) have a high support value in the phylogenetic tree in a distinct clade (Figure 1). The ITS and LSU base pair differences between *D. rhododendricola* and other related species are shown in Table 4.

*Discosia rhododendricola* KUN-HKAS 123205 is closely related to the clade consisting of *D. muscicola* CBS 109.48, *D. tricellularis* MAFF 237478, NBRC 32705, and *D. yakushimensis* MAFF 242774 (Figure 5). The results of molecular analyses based on the Genealogical Concordance Phylogenetic Species Recognition (GCPSR) also showed that *D. rhododendricola* KUN-HKAS 123205 can be distinguished as a separate species by genealogical concordance (PHI = 1.0).

#### 3.2.2. Neopestalotiopsis Rhododendricola Chaiwan & K.D. Hyde, sp. Nov. (Figure 6)

MycoBank number: 845144; Facesoffungi number: FoF 10475

Etymology: Name reflects the host from which the fungus was isolated.

Holotype: KUN-HKAS 123204

**Figure 6 jof-08-00907-f006:**
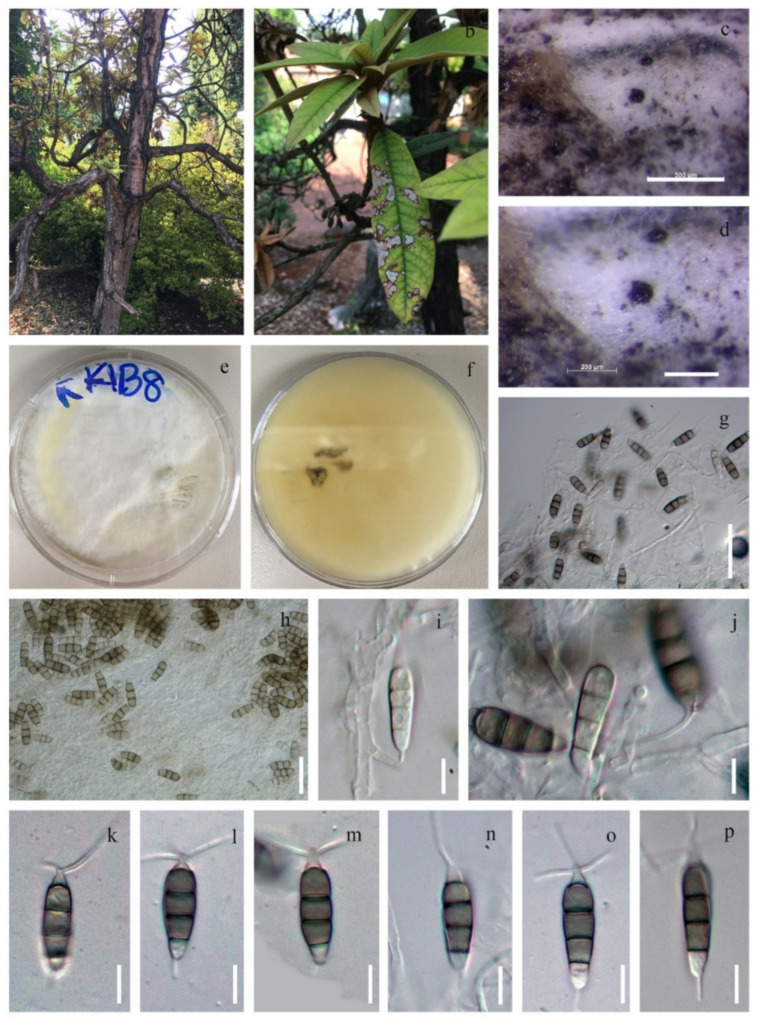
*Neopestalotiopsis rhododendricola* (KUN-HKAS 123204). (**a**) The habitat of the host plant (*Rhododendron* sp.); (**b**) Pycnidia with drops of conidial exudate on the leaf surface; (**c**,**d**) colonies growing on PDA; (**e**,**f**) culture; (**g**–**j**) conidiogenous cells and developing conidia; (**k**–**p**) conidia. Scale bars: (**c**) 500 µm; (**d**) 200 µm; (**h**) 20 µm; (**i**–**p**) 10 µm.

Saprobic on dead leaves of *Rhododendron* sp. **Sexual morph**: Undetermined. **Asexual morph**: Conidiomata (on PDA) 60–80 × 50–75 μm, pycnidial, cervular, applanate to disc-like, partly immersed or superficial, globose to clavate, solitary or confluent, embedded or semi-immersed to erumpent, dark brown, exuding globose, dark brown to black conidial masses, rounded to irregular in outline, glabrous, and unilocular or divided into several locules by tissue cells. Conidiophores are indistinct, arising from the base, hyaline, filiform to cylindrical, smooth, and are often reduced to conidiogenous cells. Conidiogenous cells appeared subcylindrical, flaskshaped, hyaline, smooth, and phialidic, with each producing a single conidium. Conidia 20–30 × 5–7 μm ($\bar{x}$ = 25 × 6 μm, n = 30), subcylindrical fusoid, ellipsoid, straight to slightly curved, 4-septate, (19–28) × 5–7 μm ($\bar{x}$ = 23.5 × 6 μm, n = 30), μm; basal cell conic with a truncate base, hyaline, rugose and thin-walled, with constrictions at the septa, hyaline, smooth-walled; with a long, tubular base, two median cells subcylindrical, second cell joined to the base, 10–15 μm ($\bar{x}$ = 12.5 μm) long, the third cell joined to the apex, 11–15 μm ($\bar{x}$ = 13 μm) long; apical cell subconical with a rounded apex; apical and basal cells each with a subapical, unbranched, filiform, straight appendage; apical appendage, 9–11 μm ($\bar{x}$ = 10 μm), basal appendage, 20–25 μm ($\bar{x}$ = 22.5 μm).

Culture characteristics: Colonies grown on PDA, with an undulating edge, reached 4–5 cm in 5 days at 25 °C, mycelium superficial, branched, septate, white mycelium with aerial on the surface, and produced black spore mass.

Material examined: CHINA, Kunming Yunnan Province; on dead leaves of *Rhododendron* sp. (Ericaceae), 28 July 2018, Napalai Chaiwan, KIB008 (KUN-HKAS 123204, holotype; isolate MFLU20-0046; Ex-type living culture KUNCC22-10802; isolate MFLUCC22-0004).

Notes: *Neopestalotiopsis rhododendricola* (KUN-HKAS 123204 and MFLU20-0046) were isolated from a *Rhododendron* sp. in China. In the phylogenetic analyses, *N. rhododendricola* forms a distinct highly supported lineage sister to *N. sonneratae* (MFLUCC17-1745T, MFLUCC17-1744), *N. coffeae-arabicae* (HGUP4019T, HGUP4015), *N. thailandica* (MFLUCC17-1730T, MFLUCC17-1731) and *N. macadamiae* (Figure 2). *Neopestalotiopsis sonneratae* was reported from leaf spots on *Sonneronata alba* in Thailand [21], *Neopestalotiopsis thailandica* was reported from leaf spots on *Rhizophora mucronata* Lam. in Thailand [21], and *N. coffeae-arabicae* was found on leaves of *Coffea arabica* in China [75].

*Neopestalotiopsis rhododendricola* sp. nov. resembles *N. thailandica* in having similar conidial size [21], but the difference is that *N. rhododendricola* has two to three tubular appendages on the apical cell, while *N. thailandica* showed only one to two tubular appendages on the apical cell. Comparison of ITS sequence differences revealed 2 base pairs, comparison of TEF sequence differences revealed 15 base pairs, and comparison of TUB differences revealed 6 base pairs of *N. rhododendricola* and *N. thailandica*. Therefore, based on morphology and phylogeny, we justify the description of *N. rhododendricola* as a new species in the *Neopestalotiopsis* genus.

#### 3.2.3. *Diaporthe nobilis* Tanaka & S. Endô, in Endô, J. Pl. Prot. Japan 13: (1927) (Figure 7)

Faces of Fungi number: FoF 02717

Saprobic on dead leaves of *Rhododendron* sp. **Sexual morph**: Undetermined. **Asexual morph**: Conidiomata pycnidial 50–100 × 25–75 μm. ($\bar{x}$ = 75 × 50 μm, n = 10), globose to stromatic, multilocular, dark brown to black, scattered. Conidiophores were observed arising from the base, hyaline, filiform to cylindrical, smooth, straight. Conidiogenous cells, 35–40 × 1–2 μm ($\bar{x}$ = 37.5 × 1.5 μm, n = 10), phialidic, cylindrical, terminal and lateral, slightly tapered towards the apex, with visible periclinal, thickening, hyaline, and smooth-walled. Beta conidia 16–20 × 1–2 μm ($\bar{x}$ = 18 × 1.5 μm, n = 30), hyaline smooth, guttulate, fusoid to ellipsoid, straight, tapered towards both ends, apex sub obtuse, base sub truncate, and aseptate. Alpha conidia not found.

Culture characteristics: Colonies grew on PDA, filamentous, flattened, dense and felty, reaching 5–6 cm in 14 days at 25 °C, white to brown on the surface, mycelium superficial, branched, and septate.

**Figure 7 jof-08-00907-f007:**
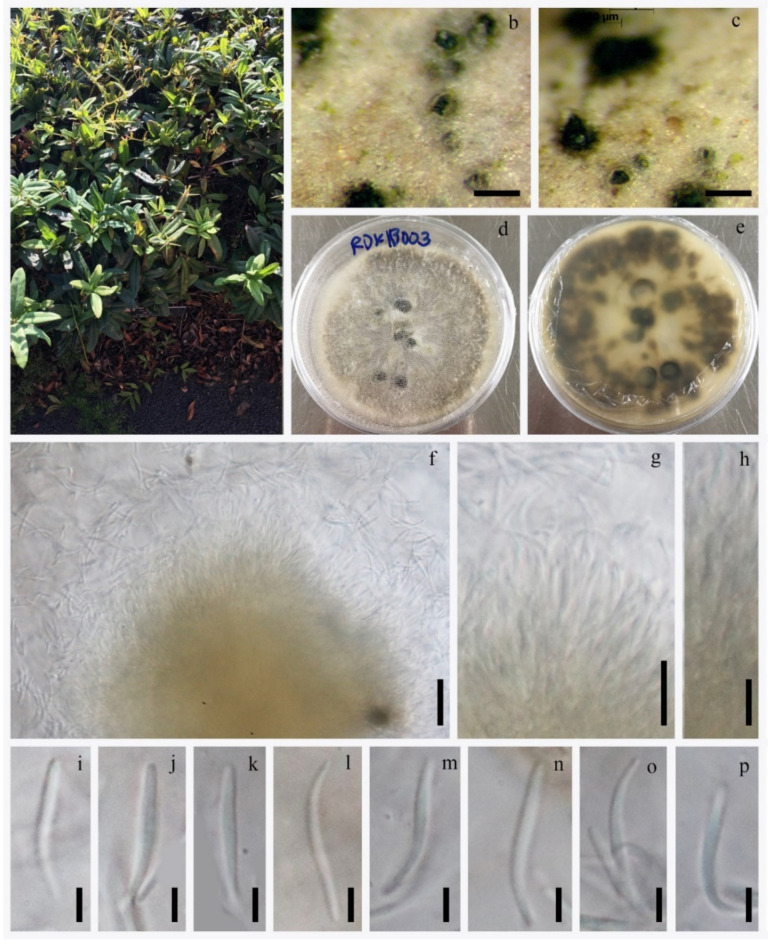
*Diaporthe nobilis* (KUN-HKAS 123203). (**a**) Habitat of host; (**b**,**c**) appearance of fungi on host surface; (**d**,**e**) culture characters on PDA; (**g**,**h**) conidiophore with attached conidium; (**i**–**p**) conidia. Scale bars: (**b**,**c**) 200 μm; (**d**–**f**,**i**–**p**) 10 μm.

Material examined: CHINA, Kunming Yunnan Province, on dead leaves of *Rhododendron* sp. (Ericaceae), 28 July 2018, Napalai Chaiwan, KIB003 (KUN-HKAS 123203, new host record; isolate MFLU20-0485; living culture KUNCC22-10803; isolate MFLUCC 18–1482.

Notes: *Diaporthe nobilis* KUN-HKAS 123203 clustered with *D. nobilis* CBS 587.79 and CBS113470 with high 0.99 PP bootstrap support. Conidiomata from the MFLUCC 18–1482 strain was acervular, semi-immersed, globose to eustromatic, and multilocular, while *D. nobilis* CBS 587.79 has pycnidia subcuticular, scattered to confluent, and uniloculate. Our strain was observed to share similar morphological characteristics with other *Diaporthe nobilis* strains in having conidiogenous cells formed at the apex of the conidiophores, cylindric, straight or curved hyaline and smooth-walled. Comparison of ITS, TEF1 and TUB2 sequence data of isolate KUN-HKAS 123203 and *D. nobilis* CBS113470, revealed 9 bp (1.41%) in 637 ITS (+5.8S) nucleotides, 2 bp (0.40%) in 496 TEF1 nucleotides and 6 bp (0.71%) in 844 TUB2 nucleotides. Therefore, we consider our strain (KUN-HKAS 123203) as *D. nobilis* and as a new host record from *Rhododendron* sp. in China.

## 4. Discussion

*Discosia* species are distributed on various vascular plants and a wide range of hosts, and occur primarily in their asexual state as endophytes, saprobes and pathogens [20,58]. Host-specificity of species in this genus has not yet been established. *Discosia* species can be found on *Fagus sylvatica* (*Fagaceae*), *Gaultheria procumbens* (*Ericaceae*), *Platanus orientalis* (*Platanaceae*), *Quercus* sp. (*Fagaceae*), *Syzygium cumini* (*Myrtaceae*), *Smilax rotundifolia* (*Smilacaceae*), and leaves of undetermined plants [60]. *Discosia blumencronii Bubák* was reported from *Rhododendron poniicum* [92], while other species can be found on leaves of *Beilschmeidia tarairi* (*Lauraceae*), *Brachychiton populneus* (*Malvaceae*), *Ceanothus fiedleri* (*Rhamnaceae*), *Eucalyptus* sp. (*Myrtaceae*), *Laurus nobilis* (*Lauraceae*) and *Phillyrea latifolia* (*Oleaceae*) [9,60]. *Discosia* species is distributed in temperate regions, being previously reported in Algeria, Austria, Brazil, France, Germany, India, Italy, New Zealand, Portugal, the USA, Sweden, Tunisia and Turkey [9].

The new taxon, *D. rhododendricola*, was phylogenetically related to *D. muscicola*, described by Nicot-Toulouse Morelet (1968), and isolated from *Cephalozia bicuspidate* (*Cephaloziaceae*) in France. However, no morphological data are available for comparison [94]. *Discosia rhododendricola* sp. nov. was isolated from *Rhododendron* sp. and its morphology was compared. The ascomata and conidia of *D. rhododendricola* were larger than those of *D. artocreas*, whereas the sizes of conidiophores, conidiogenous cells and apical appendage were similar.

*Discosia rhododendricola* is similar to *D. macrozamiae* (CPC 32109) [62], but the phylogenetic tree showed that our species was closely related to *D. muscicola* CBS 109.48. However, for *D. muscicola* CBS 109.48, only rDNA sequence data were available (Figure 1). It should be pointed out that when the ITS DNA sequences of *Discosia muscicola* were subjected to a blast search, the closest hits were *Aspergillus* species similar to *A. avenaceus*. Our novel species have DNA sequence data from three regions (LSU, ITS, and RPB2), but we can only compare the LSU region for *D. muscicola* CBS 109.48, as there are no sequence data of the protein coding gene available for comparison. Based on the previous study of Wijayawardene et al. [16], 34 genera are recognized in Sporocadaceae. In this study, we introduce *Discosia rhododendricola* as a new species based on phylogenetic analyses and the pairwise homoplasy index.

*Discosia* species share similar morphological characters, but most characters are not meaningful in species delineation. In this study, our new species constitutes a different branching pattern in our phylogeny and appears distinct from extant species. A relationship among species based on similar conidial characters does not necessarily correlate with our phylogenetic relationships, and this indicates that morphology has little significance for reliable species identification.

Herein, we introduce a new species, *Neopestalotiopsis rhododendricola* KUN-HKAS 123204, within the *Neopestalotiopsis* genus that was separated from the other *Neopestalotiopsis* clade based on morphological and molecular phylogenetic analyses (Figure 2). *Neopestalotiopsis* are characterized by their conidia with versicolor median cells, by indistinct conidiogenous cells [15] and the ITS, TUB2 and TEF1 sequences. The newly described species is phylogenetically related to the group of *N. sonneratae*, *N. coffeae-arabicae* and *N. thailandica* in the phylogenetic tree (Figure 2), and the relationship is not strongly supported. Our new species was found on a *Rhododendron* sp. plant host from China, while *N. sonneratae* was reported on leaf spots on *Sonneronata alba* L. [21] and *Neopestalotiopsis thailandica* was reported on leaf spots of *Rhizophora mucronata* Lam. Both strains have been reported in Thailand [21], and *N. coffeae-arabicae* was found on leaves of *Coffea arabica* in China [75].

*Diaporthe* species have been reported as plant pathogens, saprobes and endophytes on many plant hosts [23,28,58]. Species of *Diaporthe* are not host-specific [6,28,40]. Substrates colonized by members of *Diaporthe* recorded to date are mainly dicotyledons of Ericaceae, Fagaceae, Pinaceae, Rhizophoraceae, Rosaceae and Theaceae. Some species of *Diaporthe* can be found on more than one host. For example, *Diaporthe nobilis* was reported on *Camellia sinensis* (Theaceae), *Castanea sativa* (Fagaceae), *Malus pumila* (Rosaceae), *Pinus pantepella* (Pinaceae), *Pyrus pyrifolia* (Rosaceae) and *Rhododendron* sp. (Ericaceae) [6,28,40,58]. *Diaporthe* is mostly presented in the asexual morph as coelomycetes [23]. *Diaporthe nobilis* complex [6] has alpha and beta conidia [28]. However, our strain was only found to have beta conidia.

*Diaporthe* have been reported on *Rhododendron* spp. from Europe (Latvia) [6]. The strain (KUN-HKAS 123203) was isolated from Asia (China), indicating that the species is distributed in different geographical locations on the host; however, there is a need for more collections of microfungi associated with *Rhododendron*, targeting a wide variety of geographical locations. A checklist for *Discosia* species associated with *Rhododendron* is also provided herein.

## 5. *Discosia* Species Associated with *Rhododendron* sp.: Habitat, Known Distribution and Checklist

The above information is based on the USDA Systematic Mycology and Microbiology Laboratory (SMML) database [58], relevant literature, date from this study while current names and fungal classifications used are according to Index Fungorum (2022) [14], and an outline of Ascomycota [16]. Species confirmed with DNA sequence data are marked with an asterisk.

*Discosia artocreas* (Tode) Fr., Summa veg. Scand., Sectio Post. (Stockholm): 423 (1849)= *Sphaeria artocreas* Tode, F. Meckl. 2: 77, 1791; Fries, Syst. Myc. 2: 523, 1823.Habitat: *Rhododendron arboretum*, *R. campylocarpum*, *R. nudiflorum* [95,96], *R. catawbiense*, *R. maximum* [97], *R. ponticum* [98,99] and *Rhododendron* sp. [95,96,100]Known distribution: Italy [98], Maryland [95,96,97], New York [97], United Kingdom [100], Turkey* [99], Washington [95,96].*Discosia blumencronii* Bubák, in Handel-Mazzetti, Annln K. K. naturh. Hofmus. Wien 23: 106 (1910)Habitat: *Rhododendron ponticum* (on dead leaves) [101]Known distribution: Turkey [101]*Discosia himalayensis* Died., Annls mycol. 14(3/4): 218 (1916)= Discosia *strobilina* Lib. ex Sacc., Syll. Fung. (Abellini) 3: 656 (1884)Habitat: *Rhododendron arboretum*, *R. campanulatum* (on dead leaves) [101,102,103]Known distribution: India [101,102,103]*Discosia rhododendri* (Speschnew, Monit. Jard. Bot. Tiflis 4: 10 (1906)Habitat: *Rhododendron albrechtii* (on dead leaves)* [104], *R. ponticum** [99], *Rhododendron* sp. (on leaves) [101]Known distribution: Japan* [104], Turkey* [99]*Discosia rhododendricola* (This study *)Habitat: *Rhododendron* sp. (on dead leaves) (This study *)Known distribution: China (This study *)*Discosia* sp.Habitat: *Rhododendron* sp.* [104]Known distribution: Japan* [104]*Discosia tricellularis* (Okane, Nakagiri & Tad. Ito) F. Liu, L. Cai & Crous, in Liu, Bonthond, Groenewald, Cai & Crous, Stud. Mycol. 92: 322 (2018) (2019)Habitat: *Rhododendron indicum* [105]Known distribution: Japan [105]*Discosia vagans* De Not., Atti Acad. Tor.: 354 (1849)Habitat: *Rhododendron arboretum*, *R. nilagiricum*, *R. veitchianum** [59,103], R. ponticum* [61]Known distribution: India* [99,105], Scotland* [59]

## Figures and Tables

**Figure 1 jof-08-00907-f001:**
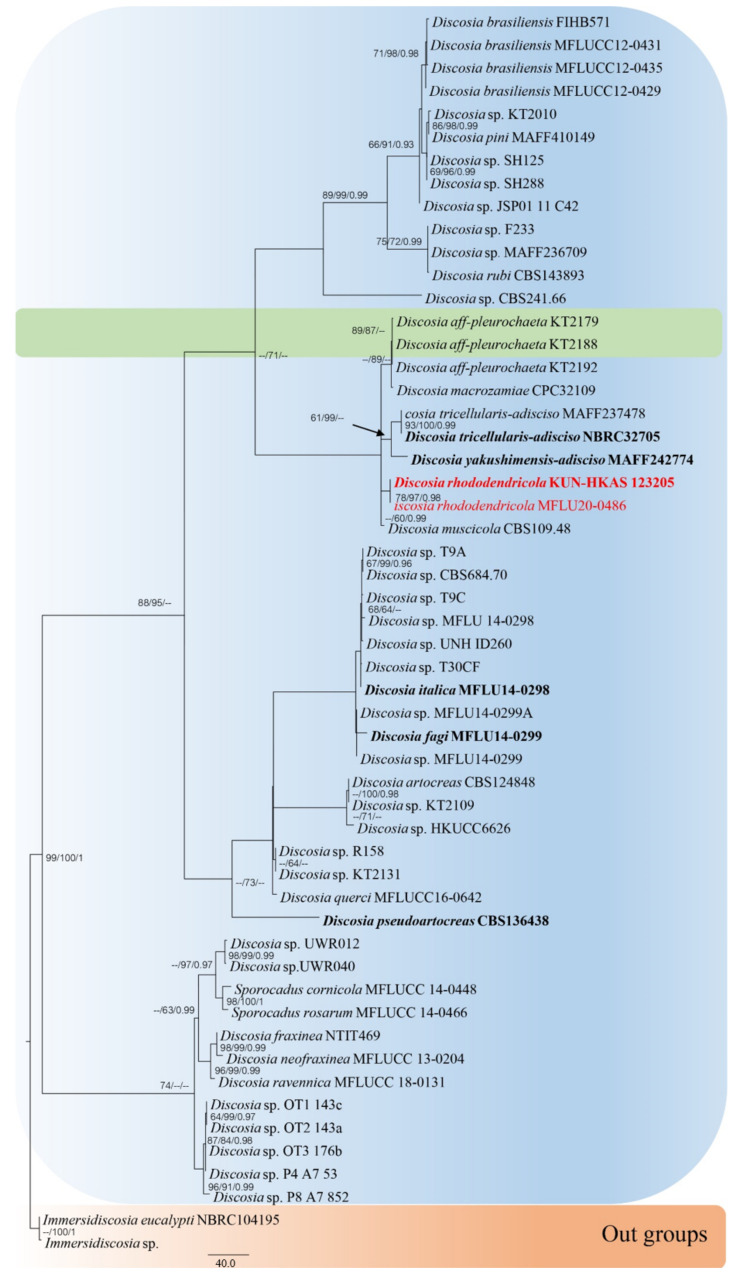
Phylogram generated from maximum parsimony analysis of LSU, ITS, RPB2, TEF1 and TUB2 gene regions. Bootstrap support values for MP and ML equal to or greater than 60% and Bayesian posterior probabilities (PP) equal to or greater than 0.90 are defined as MP/ML/PP above or below the nodes. Taxonomic novelty is indicated in red. The tree is rooted with *Immersidiscosia eucalypti* (MFLU 16-1372) and (NBRC 104195).

**Figure 2 jof-08-00907-f002:**
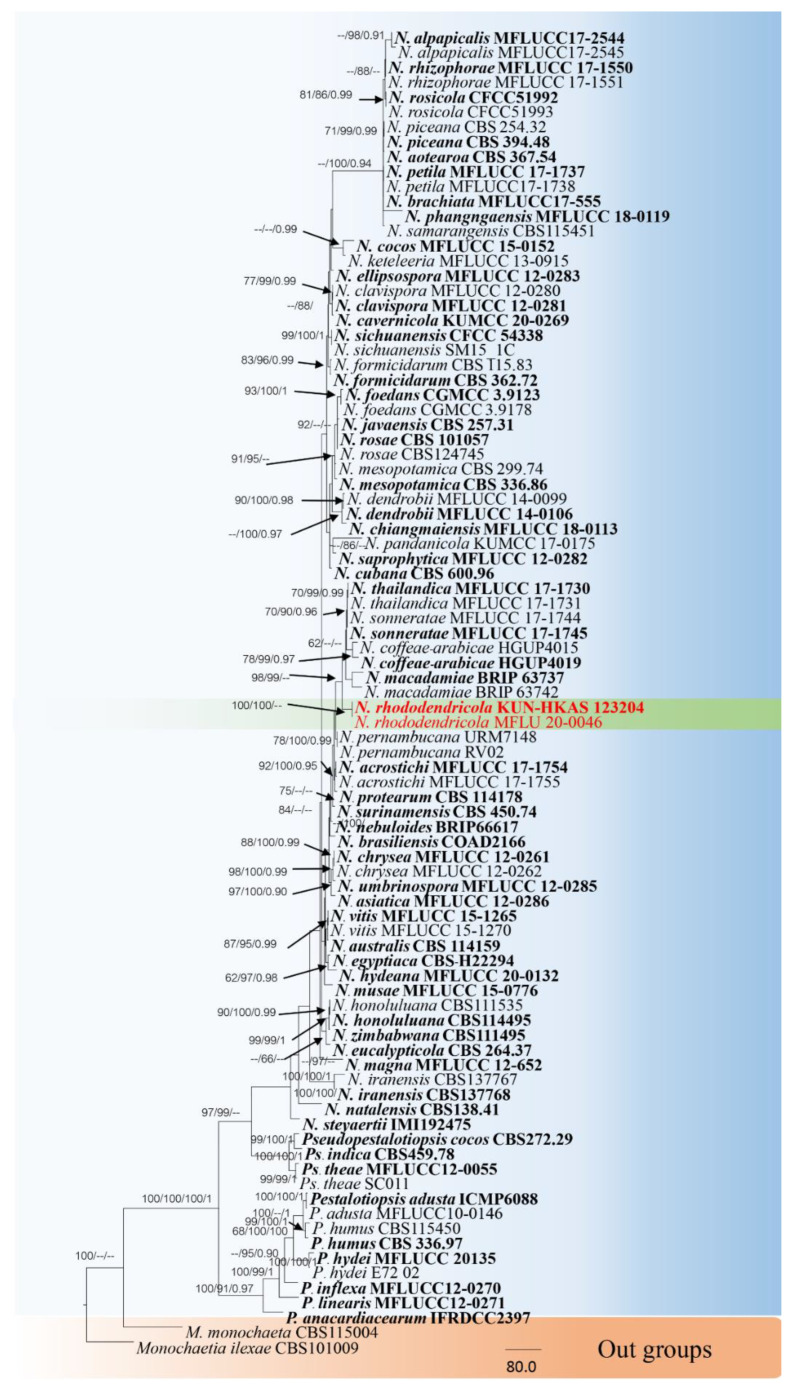
RAxML tree based on a combined dataset of ITS, TUB2 and TEF1 gene regions. Bootstrap support values for ML and MP equal to or greater than 60% and Bayesian posterior probabilities (PP) equal to or greater than 0.90 are defined as ML/MP/PP above or below the nodes. Our new taxon is indicated in red. The tree was rooted with *Monochaetia monochaeta* strains (CBS115004) and *Monochaetia ilexae* strains (CBS101009).

**Figure 3 jof-08-00907-f003:**
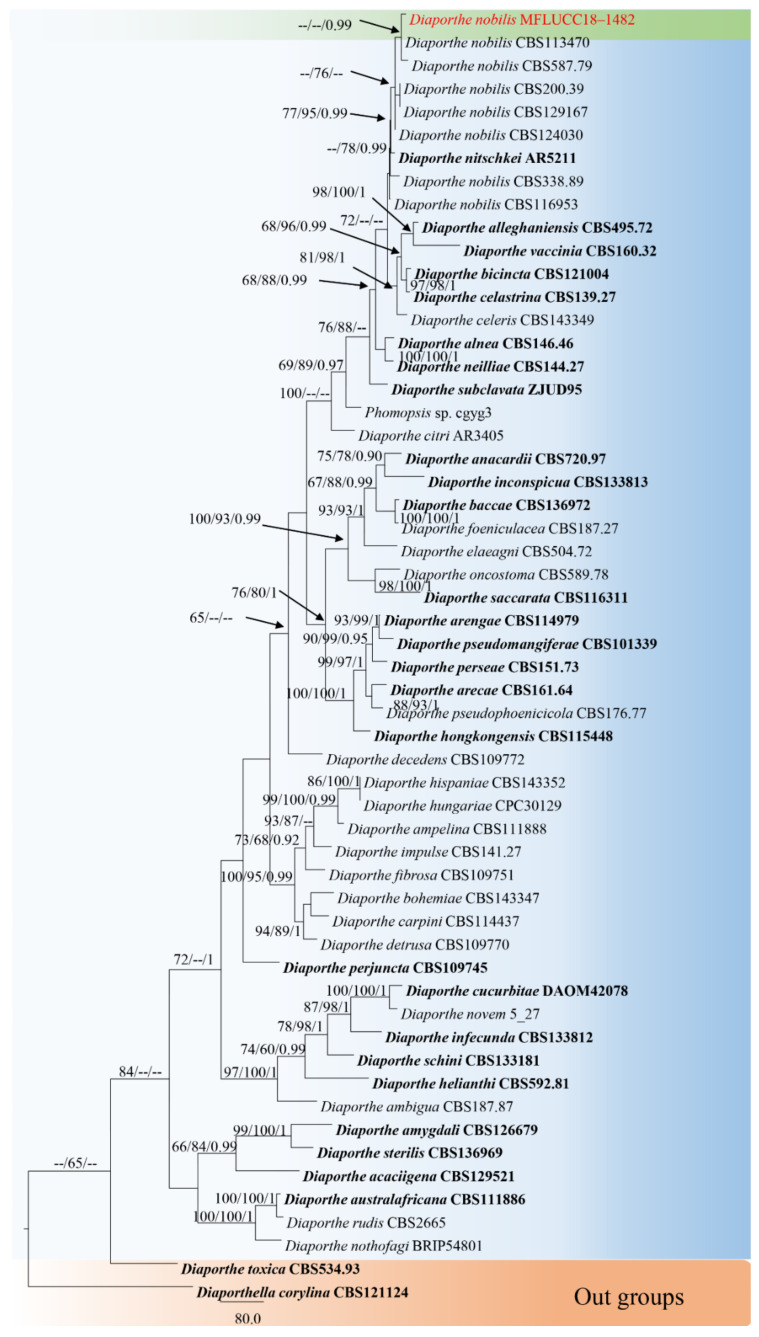
RAxML tree based on a combined dataset of ITS, TEF1 and TUB2 gene regions. Bootstrap support values for ML and MP equal to or greater than 60% and Bayesian posterior probabilities (PP) equal to or greater than 0.90 are defined as ML/MP/PP above or below the nodes. Our new taxon is indicated in red. The tree was rooted with *Diaporthella corylina* (CBS 121124).

**Figure 5 jof-08-00907-f005:**
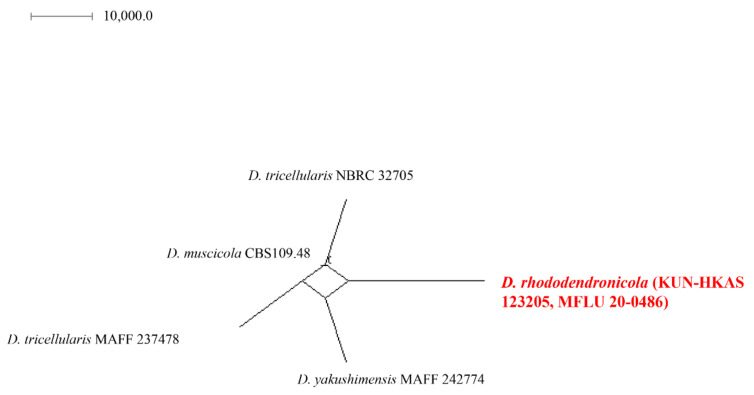
Results of the pairwise homoplasy index (PHI) test of closely related species using both LogDet transformation and splits decomposition. PHI test results (Φw) <0.05 indicate significant recombination within the dataset. The new taxon is in red bold type.

**Table 1 jof-08-00907-t001:** Culture collection numbers and GenBank accession numbers for *Discosia* used in this study. The type species are indicated in bold. The newly generated sequences are indicated in red. Instances where the GenBank Accession No. did not show the molecular data are marked with a dash.

Species Name	Culture Collection No.	Substrate/Host	Country	GenBank Accession No					References
				LSU	ITS	TUB2	TEF1	RPB2	
*Discosia pleurochaeta*	KT 2179	-	-	KT281912	KT284775	-	-	-	[9]
*Discosia pleurochaeta*	KT 2188	-	-	AB593713	AB594781	AB594179	-	-	[9]
*Discosia pleurochaeta*	KT 2192	-	-	AB593714	AB594782	AB594180	-	-	[9]
*Discosia artocreas*	CBS 124848	Fagus sylvatica	Germany	MH554213	MH553994	MH554662	MH554420	MH554903	[13]
*Discosia brasiliensis*	MFLUCC 12-0429	Dead leaf	Thailand	KF827436	KF827432	KF827469	KF827465	KF827473	[59]
*Discosia brasiliensis*	MFLUCC 12-0431	Dead leaf	Thailand	KF827437	KF827433	KF827470	KF827466	KF827474	[59]
*Discosia brasiliensis*	MFLUCC 12-0435	Dead leaf	Thailand	KF827438	KF827434	KF827471	KF827467	KF827475	[59]
*Discosia fagi*	MFLU 14-0299A	Fagus sylvatica	Italy	KM678048	KM678040	-	-	-	[60]
*Discosia fagi*	MFLU 14-0299B	Fagus sylvatica	Italy	KM678047	KM678039	-	-	-	[60]
** *Discosia fagi* **	**MFLU 14-0299C**	**Fagus sylvatica**	**Italy**	**KM678048**	**KM678040**	**-**	**-**	**-**	**[60]**
** *Discosia italica* **	**MFLU 14-0298B**	**Fagus sylvatica**	**Italy**	**KM678045**	**KM678042**	**-**	**-**	**-**	**[60]**
*Discosia italica*	MFLU 14-0298C	Fagus sylvatica	Italy	KM678044	KM678041	-	-	-	[60]
*Discosia macrozamiae*	CPC 32109	-	-	MH327856	MH327820	MH327895	MH327884	-	[61]
*Discosia muscicola*	CBS 109.48	-	-	MH867828	-	-	-	-	[62]
*Discosia neofraxinea*	NTIT 469	Fagus sylvatica	Italy	KF827439	KF827435	KF827472	KF827468	KF827476	[59]
*Discosia neofraxinea*	MFLUCC 13-0204	Fagus sylvatica	Italy	KR072672	KR072673	-	-	-	[10]
** *Discosia pseudoartocreas* **	**CBS 136438**	**Tilia sp.**	**Austria**	**KF777214**	**KF777161**	**MH554672**	**MH554430**	**MH554913**	**[63]**
*Discosia pini*	MAFF 410149	Pinus densiflora	Japan	AB593708	AB594776	AB594174	-	-	[9]
*Discosia querci*	MFLUCC 16-0642	-	-	MG815830	MG815829	-	-	-	[64]
*Discosia ravennica*	MFLU 18-0131	Pyrus sp.	Italy	MT376617	MT376615	MT393594	-	-	[46]
** * Discosia rhododendricola * **	** KUN-HKAS 123205 **	** * Rhododendron * ** ** sp. **	** China **	** MT741963 **	** MT741959 **	** - **	** - **	** MW143037 **	** This study **
	** MFLU20-0486 **	** * Rhododendron * ** ** sp. **	** China **	** OP162409 **	** OP162414 **	** - **	** - **	** OP169687 **	** This study **
*Discosia rubi*	CBS 143893	Rubus phoenicolasius	USA	MH554334	MH554131	MH554804	MH554566	MH555038	[13]
*Discosia* sp.	F 233	-	-	-	KU751876	-	-	-	[13]
*Discosia* sp.	3T30CF	-	-	-	FJ861385	-	-	-	[65]
*Discosia* sp.	3T9A	-	-	-	FJ861386	-	-	-	[65]
*Discosia* sp.	3T9C	-	-	-	FJ861387	-	-	-	[65]
*Discosia* sp.	FIHB 571	-	-	-	DQ536523	-	-	-	[66]
*Discosia* sp.	HKUCC 6626	-	-	AF382381	AF405303	-	-	-	[67]
*Discosia* sp.	JSP0111c42	-	-	-	KR093849	-	-	-	[68]
*Discosia* sp.	KT 2193	-	-	AB593706	AB594774	-	-	-	[9]
*Discosia* sp.	OT1 143c	-	-	-	KT804147	-	-	-	[13]
*Discosia* sp.	OT2 143a	-	-	-	KT804075	-	-	-	[13]
*Discosia* sp.	OT3 176b	-	-	-	KT804146	-	-	-	[13]
*Discosia* sp.	P4 A7 53	-	-	-	KU325138	-	-	-	[13]
*Discosia* sp.	P8 A7-852	-	-	-	KU325418	-	-	-	[13]
*Discosia* sp.	R 158	-	-	-	JN689956	-	-	-	[13]
*Discosia* sp.	UNH ID260	-	-	-	KX459431	-	-	-	[13]
*Discosia* sp.	UWR 012	-	-	-	KX426948	-	-	-	[13]
*Discosia* sp.	UWR 040	-	-	-	KX426977	-	-	-	[13]
*Discosia* sp.	KT 2109	-	-	-	MT236494	-	-	-	[69]
*Discosia* sp.	SH 125	-	-	-	JF449727	-	-	-	[13]
*Discosia* sp.	SH 288	-	-	-	AB594783	-	-	-	[9]
*Discosia* sp.	MAFF 236709	-	-	-	KU751876	-	-	-	[13]
*Discosia* sp.	CBS 241.66	Acacia karroo	South Africa	MH554244	MH554022	MH554698	-	-	[13]
*Discosia* sp.	CBS 684.70	Aesculus hippocastanum	Netherlands	MH554277	MH554064	MH554740	-	-	[13]
*Discosia tricellularis*	MAFF 237478	-	-	AB593730	AB594798	AB594189	-	-	[9]
** *Discosia tricellularis* **	**NBRC 32705**	**Rhododendron indicum**	**Japan**	**AB593728**	**AB594796**	**AB594188**	**-**	**-**	**[9]**
** *Discosia yakushimensis* **	**MAFF 242774**	**Symplocos prunifolia**	**Japan**	**AB593721**	**AB594789**	**AB594187**	**-**	**-**	**[9]**
*Sporocadus cornicola*	MFLUCC 14-0448	Cornus sanguinea	Italy	-	KU974967	-	-	-	[70]
*Sporocadus rosarum*	MFLUCC 14-0466	Rosa canina	Italy	KT281912	KT284775	-	-	-	[70]

**Table 2 jof-08-00907-t002:** Culture collection numbers and GenBank accession numbers for *Neopestalotiopsis* used in this study. The type species are indicated in bold. The newly generated sequences are indicated in red. Instances where the GenBank Accession No. did not show the molecular data are marked with the dash.

Species Name	Culture Collection No.	Substrate/Host	Country	GenBank Accession No			References
				ITS	TUB2	TEF1	
*Monochaetia ilexae*	CBS 101009	Air	Japan	MH55395	MH554612	MH554371	[13]
*M. monochaeta*	CBS 115004	Quercus robur	Netherlands	AY853243	MH554639	MH554398	[13]
** *Neopestalotiopsis acrostichi* **	**MFLUCC 17-1754**	**Acrostichum aureum**	**Thailand**	**MK764272**	**MK764338**	**MK764316**	**[21]**
*N. acrostichi*	MFLUCC 17-1755	Acrostichum aureum	Thailand	MK764273	MK764339	MK764317	[21]
** *N. alpapicalis* **	**MFLUCC 17-2544**	**Rhizophora mucronata**	**Thailand**	**MK357772**	**MK463545**	**MK463547**	**[71]**
*N. alpapicalis*	MFLUCC 17-2545	Symptomatic Rhizophora apiculata leaves	Thailand	MK357773	MK463546	MK463548	[71]
** *N. aotearoa* **	**CBS 367.54**	**Canvas**	**New Zealand**	**KM199369**	**KM199454**	**KM199526**	**[15]**
** *N. asiatica* **	**MFLUCC 12-0286**	**Prunus dulcis**	**China**	**JX398983**	**JX399018**	**JX399049**	**[15]**
** *N. australis* **	**CBS 114159**	**Telopea sp.**	**Australia**	**KM199348**	**KM199432**	**KM199537**	**[15]**
** *N. brachiata* **	**MFLUCC 17-555**	**Rhizophora apiculata**	**Thailand**	**MK764274**	**MK764340**	**MK764318**	**[21]**
** *N. brasiliensis* **	**COAD 2166**	**Psidium guajava**	**Brazil**	**MG686469**	**MG692400**	**MG692402**	**[72]**
** *N. cavernicola* **	**KUMCC 20-0269**	**Cave**	**China**	**MW545802**	**MW557596**	**MW550735**	**[22]**
** *N. chiangmaiensis* **	**MFLUCC 18-0113**	**Pandanus sp.**	**Thailand**	**-**	**MH412725**	**MH388404**	**[73]**
** *N. chrysea* **	**MFLUCC 12-0261**	**Dead leaves**	**China**	**JX398985**	**JX399020**	**JX399051**	**[74]**
*N. chrysea*	MFLUCC 12-0262	Dead plant	China	JX398986	JX399021	JX399052	[74]
*N. clavispora*	MFLUCC 12-0280	Magnolia sp.	China	JX398978	JX399013	JX399044	[74]
** *N. clavispora* **	**MFLUCC 12-0281**	**Magnolia sp.**	**China**	**JX398979**	**JX399014**	**JX399045**	**[74]**
** *N. cocoës* **	**MFLUCC 15-0152**	**Cocos nucifera**	**Thailand**	**KX789687**	**-**	**KX789689**	**[17]**
*N. coffeae-arabicae*	HGUP4015	Coffea arabica	China	KF412647	KF412641	KF412644	[75]
** *N. coffeae-arabicae* **	**HGUP4019**	**Coffea arabica**	**China**	**KF412649**	**KF412643**	**KF412646**	**[75]**
** *N. cubana* **	**CBS 600.96**	**Leaf Litter**	**Cuba**	**KM199347**	**KM199438**	**KM199521**	**[15]**
*N. dendrobii*	MFLUCC 14-0099	Dendrobium cariniferum	Thailand	MK993570	MK975834	MK975828	[76]
** *N. dendrobii* **	**MFLUCC 14-0106**	**Dendrobium cariniferum**	**Thailand**	**MK993571**	**MK975835**	**MK975829**	**[76]**
** *N. egyptiaca* **	**CBS H 22294**	**Mangifera indica**	**Egypt**	**KP943747**	**KP943746**	**KP943748**	**[77]**
** *N. ellipsospora* **	**MFLUCC 12-0283**	**Dead plant materials**	**China**	**JX398980**	**JX399016**	**JX399047**	**[74]**
** *N. eucalypticola* **	**CBS 264.37**	**Eucalyptus globulus**	**-**	**KM199376**	**KM199431**	**KM199551**	**[15]**
** *N. foedans* **	**CGMCC 3.9123**	**Mangrove plant**	**China**	**JX398987**	**JX399022**	**JX399053**	**[74]**
*N. foedans*	CGMCC 3.9178	Neodypsis decaryi	China	JX398989	JX399024	JX399055	[74]
*N. formicidarum*	CBS 115.83	Plant debris	Cuba	KM199344	KM199444	KM199519	[78]
** *N. formicidarum* **	**CBS 362.72**	**Dead Formicidae (ant)**	**Cuba**	**KM199358**	**KM199455**	**KM199517**	**[78]**
*N. honoluluana*	CBS 111535	Telopea sp.	USA	KM199363	KM199461	KM199546	[15]
** *N. honoluluana* **	**CBS 114495**	**Telopea sp.**	**USA**	**KM199364**	**KM199457**	**KM199548**	**[15]**
** *N. hydeana* **	**MFLUCC 20-0132**	**Artocarpus heterophyllus**	**Thailand**	**MW266069**	**MW251119**	**MW251129**	**[79]**
*N. iranensis*	CBS 137767	Fragaria ananassa	Iran	KM074045	KM074056	KM074053	[80]
** *N. iranensis* **	**CBS 137768**	**Fragaria ananassa**	**Iran**	**KM074048**	**KM074057**	**KM074051**	**[81]**
** *N. javaensis* **	**CBS 257.31**	**Cocos nucifera**	**Java**	**KM199357**	**KM199437**	**KM199543**	**[15]**
*N. keteleeria*	MFLUCC 13-0915	Keteleeria pubescens	China	KJ023087	KJ023088	KJ023089	[75]
** *N. macadamiae* **	**BRIP 63737c**	**Macadamia integrifolia**	**Australia**	**KX186604**	**KX186654**	**KX186627**	**[81]**
*N. macadamiae*	BRIP 63742a	Macadamia integrifolia	Australia	KX186599	KX186657	KX186629	[82]
** *N. magna* **	**MFLUCC 12-652**	**Pteridium sp.**	**France**	**KF582795**	**KF582793**	**KF582791**	**[15]**
*N. mesopotamica*	CBS 299.74	Eucalyptus sp.	Turkey	KM199361	KM199435	KM199541	[15]
** *N. mesopotamica* **	**CBS 336.86**	**Pinus brutia**	**Iraq**	**KM199362**	**KM199441**	**KM199555**	**[15]**
** *N. musae* **	**MFLUCC 15-0776**	**Musa sp.**	**Thailand**	**KX789683**	**KX789686**	**KX789685**	**[17]**
** *N. natalensis* **	**CBS 138.41**	**Acacia mollissima**	**South Africa**	**KM199377**	**KM199466**	**KM199552**	**[15]**
** *N. nebuloides* **	**BRIP 66617**	**Sporobolus elongatus**	**Australia**	**MK966338**	**MK977632**	**MK977633**	**[82]**
*N. pandanicola*	KUMCC 17-0175	Pandanus sp.	China	-	MH412720	MH388389	[73]
*N. pernambucana*	URM7148	Vismia guianensis	Brazil	KJ792466	-	KU306739	[83]
*N. pernambucana*	RV02	Vismia guianensis	Brazil	KJ792467	-	KU306740	[83]
** *N. petila* **	**MFLUCC 17-1737**	**Rhizophora mucronata**	**Thailand**	**MK764275**	**MK764341**	**MK764319**	**[21]**
*N. petila*	MFLUCC 17-1738	Rhizophora mucronata	Thailand	MK764276	MK764342	MK764320	[21]
** *N. phangngaensis* **	**MFLUCC 18-0119**	**Pandanus sp.**	**Thailand**	**MH388354**	**MH412721**	**MH388390**	**[73]**
*N. piceana*	CBS 254.32	Cocos nucifera	Indonesia	KM199372	KM199452	KM199529	[15]
** *N. piceana* **	**CBS 394.48**	**Picea sp.**	**UK**	**KM199368**	**KM199453**	**KM199527**	**[15]**
** *N. protearum* **	**CBS 114178**	**Leucospermum cuneiforme cv. “Sunbird”**	**Zimbabwe**	**JN712498**	**KM199463**	**LT853201**	**[15]**
** *N. rhizophorae* **	**MFLUCC 17-1550**	**Rhizophora mucronata**	**Thailand**	**MK764277**	**MK764343**	**MK764321**	**[21]**
*N. rhizophorae*	MFLUCC 17-1551	Rhizophora mucronata	Thailand	MK764278	MK764344	MK764322	[21]
** * N. rhododendricola * **	** KUN-HKAS 123204 **	** * Rhododendron * ** ** sp. **	** China **	** OK283069 **	** OK274147 **	** OK274148 **	** This study **
	** MFLU20-0046 **	** * Rhododendron * ** ** sp. **	** China **	** OP11897554 **	** OP169689 **	** OP169688 **	** This study **
** *N. rosae* **	**CBS 101057**	**Rosa sp.**	**New Zealand**	**KM199359**	**KM199429**	**KM199523**	**[15]**
*N. rosae*	CBS 124745	Paeonia suffruticosa	USA	KM199360	KM199430	KM199524	[15]
** *N. rosicola* **	**CFCC 51992**	**Rosa chinensis**	**China**	**KY885239**	**KY885245**	**KY885243**	**[84]**
*N. rosicola*	CFCC 51993	Rosa chinensis	China	KY885240	KY885246	KY885244	[84]
*N. samarangensis*	CBS 115451	Unidentified tree	China	KM199365	KM199447	KM199556	[85]
** *N. saprophytica* **	**MFLUCC 12-0282**	**Magnolia sp.**	**China**	**JX398982**	**JX399017**	**JX399048**	**[74]**
** *N. sichuanensis* **	**CFCC 54338**	**Castanea mollissima**	**China**	**MW166231**	**MW218524**	**MW199750**	**[86]**
*N. sichuanensis*	SM15-1C	Castanea mollissima	China	MW166232	MW218525	MW199751	[86]
*N. sonneratae*	MFLUCC 17-1744	Sonneronata alba	Thailand	MK764279	MK764345	MK764323	[21]
** *N. sonneratae* **	**MFLUCC 17-1745**	**Sonneronata alba**	**Thailand**	**MK764280**	**MK764346**	**MK764324**	**[21]**
** *N. steyaertii* **	**IMI 192475**	**Eucalyptus viminalis**	**Australia**	**KF582796**	**KF582794**	**KF582792**	**[15]**
** *N. surinamensis* **	**CBS 450.74**	**Soil under Elaeis guineensis**	**Suriname**	**KM199351**	**KM199465**	**KM199518**	**[15]**
** *N. thailandica* **	**MFLUCC 17-1730**	**Rhizophora mucronata**	**Thailand**	**MK764281**	**MK764347**	**MK764325**	**[21]**
*N. thailandica*	MFLUCC 17-1731	Rhizophora mucronata	Thailand	MK764282	MK764348	MK764326	[21]
** *N. umbrinospora* **	**MFLUCC 12-0285**	**Unidentified plant**	**China**	**JX398984**	**JX399019**	**JX399050**	**[74]**
** *N. vitis* **	**MFLUCC 15-1265**	**Vitis vinifera cv. “Summer black”**	**China**	**KU140694**	**KU140685**	**KU140676**	**[87]**
*N. vitis*	MFLUCC 15-1270	Vitis vinifera cv. “Kyoho”	China	KU140699	KU140690	KU140681	[87]
** *N. zimbabwana* **	**CBS 111495**	**Leucospermum cunciforme**	**Zimbabwe**	**JX556231**	**KM199456**	**KM199545**	**[15]**
** *Pestalotiopsis adusta* **	**ICMP6088**	**On refrigerator door PVC gasket**	**Fiji**	**JX399006**	**JX399037**	**JX399070**	**[74]**
*P. adusta*	MFLUCC10-0146	*Syzygium* sp.	Thailand	JX399007	JX399038	JX399071	[74]
** *P. anacardiacearum* **	**IFRDCC 2397**	**Mangifera indica**	**China**	**KC247154**	**KC247155**	**KC247156**	**[88]**
*P. humus*	CBS 115450	Ilex cinerea	China	KM199319	KM199418	KM199487	[15]
** *P. humus* **	**CBS 336.97**	**Soil**	**Papua New Guinea**	**KM199317**	**KM199420**	**KM199484**	**[15]**
** *P. hydei* **	**MFLUCC 20135**	**Litsea petiolata**	**Thailand**	**MW266063**	**MW251112**	**MW251113**	**[79]**
** *N. thailandica* **	**MFLUCC 17-1730**	**Rhizophora mucronata**	**Thailand**	**MK764281**	**MK764347**	**MK764325**	**[21]**
*N. thailandica*	MFLUCC 17-1731	Rhizophora mucronata	Thailand	MK764282	MK764348	MK764326	[21]
** *N. umbrinospora* **	**MFLUCC 12-0285**	**Unidentified plant**	**China**	**JX398984**	**JX399019**	**JX399050**	**[74]**
*P. hydei*	E-72-02	Eucalyptus grandis	Brazil	KU926708	KU926716	KU926712	[79]
** *P. inflexa* **	**MFLUCC12-0270**	**Unidentifified tree**	**China**	**JX399008**	**JX399039**	**JX399072**	**[74]**
** *P. linearis* **	**MFLUCC12-0271**	** *Trachelospermum* ** **sp.**	**China**	**JX398992**	**JX399027**	**JX399058**	**[74]**
** *Pseudopestalotiopsis cocos* **	**CBS 272.29**	**Cocos nucifera**	**Indonesia**	**KM199378**	**KM199467**	**KM199553**	**[15]**
** *Ps. indica* **	**CBS 459.78**	**Hibiscus rosa-sinensis**	**India**	**KM199381**	**KM199470**	**KM199560**	**[15]**
** *Ps. theae* **	**MFLUCC12-0055 T**	**Camellia sinensis**	**Thailand**	**JQ683727**	**JQ683711**	**JQ683743**	**[15]**
*Ps. theae*	SC011	Camellia sinensis	Thailand	JQ683726	JQ683710	JQ683742	[15]

**Table 3 jof-08-00907-t003:** Culture collection numbers and GenBank accession numbers for *Diaporthe* used in this study. The type species are indicated in bold. The newly generated sequences are indicated in red. Instances where the GenBank Accession No. did not show the molecular data are marked with the dash.

Species Name	Culture Collection No.	Substrate/Host	Country	GenBank Accession No			References
				ITS	TUB2	TEF1	
** *Diaporthe acaciigena* **	**CBS 129521**	** *Acacia retinodes* **	**-**	**KC343005**	**KC343973**	**KC343731**	**[6]**
** *Diaporthe alleghaniensis* **	**CBS 495.72**	** *Betula alleghaniensis* **	**-**	**KC343007**	**KC343975**	**KC343733**	**[6]**
** *Diaporthe alnea* **	**CBS 146.46**	***Alnus* sp.**	**-**	**KC343008**	**KC343976**	**KC343734**	**[6]**
*Diaporthe ambigua*	CBS 187.87	*Helianthus annuus*	Italy	KC343015	KC343983	KC343741	[6]
*Diaporthe ampelina*	CBS 111888	*Vaccinium vinifera*	USA	KC343016	KC343984	KC343742	[6]
** *Diaporthe amygdali* **	**CBS 126679**	** *Prunus dulcis* **	**-**	**KC343022**	**KC343990**	**AY343748**	**[6]**
** *Diaporthe anacardii* **	**CBS 720.97**	** *Anacardium ocidentale* **	**-**	**KC343024**	**KC343992**	**KC343750**	**[6]**
** *Diaporthe arecae* **	**CBS 161.64**	** *Areca catechu* **	**-**	**KC343032**	**KC344000**	**KC343758**	**[6]**
** *Diaporthe arengae* **	**CBS 114979**	** *Arenga engleri* **	**-**	**KC343034**	**KC344002**	**KC343760**	**[6]**
** *Diaporthe australafricana* **	**CBS 111886**	** *Vaccinium vinifera* **	**Australia**	**KC343038**	**KC344006**	**KC343764**	**[6]**
** *Diaporthe baccae* **	**CBS 136972**	** *Vaccinium corymbosum* **	**-**	**KJ160565**	**-**	**KJ160597**	**[45]**
** *Diaporthe bicincta* **	**CBS 121004**	***Juglans* sp.**	**-**	**KC343134**	**KC344102**	**KC343860**	**[6]**
*Diaporthe bohemiae*	CBS 143347	*Vitis* spp.	Czech Republic	MG281015	MG281188	MG281536	[29]
*Diaporthe carpini*	CBS 114437	*Carpinus betulus*	Sweden	KC343044	KC344012	KC343770	[6]
** *Diaporthe celastrina* **	**CBS 139.27**	** *Celastrus scandens* **	**-**	**KC343047**	**KC344015**	**KC343773**	**[6]**
*Diaporthe celeris*	CBS 143349	*Vaccinium vinifera*	UK	MG281017	MG281190	MG281538	[29]
** *Diaporthella corylina* **	**CBS 121124**	***Corylus* sp.**	**-**	**KC343004**	**KC343972**	**KC343730**	**[6]**
*Diaporthe citri*	AR 3405	*-*	-	KC843311	KC843187	KC843071	[89]
** *Diaporthe cucurbitae* **	**DAOM 42078**	** *Cucumis sativus* **	**-**	**KM453210**	**KP118848**	**KM453211**	**[89]**
*Diaporthe decedens*	CBS 109772	*Corylus avellana*	Austria	KC343059	KC344027	KC343785	[6]
*Diaporthe detrusa*	CBS 109770	*Berberis vulgaris*	Austria	KC343061	KC344029	KC343787	[6]
*Diaporthe elaeagni*	CBS 504.72	*Eleagnus* sp.	Netherlands	KC343064	KC344032	KC343790	[6]
** * Diaporthe nobilis * **	** KUN-HKAS 123203 **	** * Rhododendron * sp. **	** China **	** MT741962 **	** MW150988 **	** MW248138 **	** This study **
Diaporthe nobilis	CBS 338.89	*Hedera helix*	-	KC343152	KC344120	KC343878	[6]
**Diaporthe nobilis**	**CBS 200.39**	** *Laurus nobilis* **	**Germany**	**KC343151**	**KC344119**	**KC343877**	**[6]**
Diaporthe nobilis	CBS 113470	*-*	-	KC343146	-	-	[6]
Diaporthe nobilis	CBS 116953	*-*	-	KC343147	-	-	[6]
Diaporthe nobilis	CBS 124030	*-*	-	KC343149	-	-	[6]
Diaporthe nobilis	CBS 129167	*-*	-	KC343150	-	-	[6]
Diaporthe nobilis	CBS 587.79	*Pinus pantepella*	-	KC343153	KC344121	KC343879	[6]
Diaporthe fibrosa	CBS 109751	*-*	-	KC343099	KC344067	KC343825	[6]
Diaporthe foeniculacea	CBS 187.27	*-*	-	KC343107	KC344075	KC343833	[6]
**Diaporthe helianthi**	**CBS 592.81**	** *Helianthus annuus* **	**-**	**KC343115**	**KC344083**	**KC343841**	**[6]**
** *Diaporthe nitschkei* **	**AR 5211**	** *Hedera helix* **	**-**	**KJ210538**	**KJ420828**	**KJ210559**	**[89]**
*Diaporthe hispaniae*	CBS 143351	-	-	MG281124	MG281296	MG281645	[29]
** *Diaporthe hongkongensis* **	**CBS 115448**	** *Dichroa febrífuga* **	**-**	**KC343119**	**KC344087**	**KC343845**	**[6]**
*Diaporthe hungariae*	CPC 30129	*-*	-	-	-	MG281646	[29]
*Diaporthe impulse*	CBS 114434	*-*	-	KC343122	KC344089	KC343847	[6]
** *Diaporthe inconspicua* **	**CBS 133813**	** *Maytenus ilicifolia* **	**-**	**KC343123**	**KC344091**	**KC343849**	**[6]**
** *Diaporthe infecunda* **	**CBS 133812**	** *Schinus terebinthifolius* **	**-**	**KC343126**	**KC344094**	**KC343852**	**[6]**
** *Diaporthe neilliae* **	**CBS 144. 27**	***Spiraea* sp.**	**-**	**KC343144**	**KC344112**	**KC343870**	**[90]**
** *Diaporthe nothofagi* **	**BRIP 54801**	** *Nothofagus cunninghamii* **	**-**	**JX862530**	**KF170922**	**JX862536**	**[91]**
*Diaporthe novem*	CBS 127271	*-*	-	HM347710	-	HM347698	[6]
*Diaporthe oncostoma*	CBS 589.78	*-*	-	KC343162	KC344130	KC343888	[6]
** *Diaporthe perjuncta* **	**CBS 109745**	** *Ulmus glabra* **	**-**	**KC343172**	**KC344140**	**KC343898**	**[6]**
** *Diaporthe perseae* **	**CBS 151.73**	** *Persea gratissima* **	**-**	**KC343173**	**KC344141**	**KC343899**	**[6]**
** *Diaporthe pseudomangiferae* **	**CBS 101339**	** *Mangifera indica* **	**-**	**KC343181**	**KC344149**	**KC343907**	**[6]**
*Diaporthe pseudophoenicicola*	CBS 462.69	*Phoenix dactylifera*	-	KC343183	KC344151	KC343909	[6]
*Diaporthe rudis*	CBS 2665	*-*	-	-	KM396309	KM396311	[6]
** *Diaporthe saccarata* **	**CBS 116311**	** *Protea repens* **	**-**	**KC343190**	**KC344158**	**KC343916**	**[6]**
** *Diaporthe schini* **	**CBS 133181**	** *Schinus terebinthifolius* **	**-**	**KC343191**	**KC344159**	**KC343917**	**[6]**
** *Diaporthe sterilis* **	**CBS 136969**	** *Vaccinium corymbosum* **	**-**	**KJ160579**	**KJ160528**	**KJ160611**	**[92]**
** *Diaporthe subclavata* **	**ZJUD 95**	** *-* **	**-**	**KJ490630**	**KJ490451**	**KJ490509**	**[93]**
** *Diaporthe toxica* **	**CBS 534.93**	** *Lupinus angustifolius* **	**-**	**KC343220**	**KC344188**	**KC343946**	**[6]**
** *Diaporthe vaccinii* **	**CBS 160.32**	** *Vaccinium macrocarpon* **	**-**	**AF317578**	**JX270436**	**GQ250326**	**[92]**
*Phomopsis* sp.	FH 2012b	-	-	JQ954649	-	JQ954667	[93]

**Table 4 jof-08-00907-t004:** LSU and ITS nucleotides comparisons of Discosia species related to our new taxon.

	LSU								ITS								
	Base Pair Positions								Base Pair Positions								
	70	369	379	407	502	646	872	873	939	959	1003	1004	1019	1039	1303	1402	1404
*D. rhododendricola* (KUN-HKAS 123205)	G	A	T	T	G	A	C	T	C	T	A	C	A	T	A	C	T
*D. macrozamiae* CPC 32109	A	A	T	T	G	G	-	-	C	G	G	T	T	T	A	G	A
*D. muscicola* CBS 109.48	G	G	T	C	A	A	-	-	-	-	-	-	-	-	-	-	-
*D. pleurochaeta* KT2179	A	A	C	T	G	G	T	C	C	G	G	T	T	T	A	G	A
*D. pleurochae* KT 2188	A	A	T	T	G	G	T	C	C	G	G	T	T	T	A	G	A
*D. pleurochae* KT 2192	A	A	T	T	G	G	T	C	C	G	G	T	T	T	A	G	A
*D. tricellularis* MAFF237478	-	-	-	-	-	-	-	-	G	G	G	T	T	A	G	G	A
*D. tricellularis* NCBR32705	-	-	-	-	-	-	-	-	G	G	G	T	T	A	G	G	A
*D. yakushimensis* MAFF 242774	-	-	-	-	-	-	-	-	C	G	G	T	T	T	A	G	A

## Data Availability

All sequences generated in this study were submitted to GenBank.

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
