# Peer review of "Fungal Species from Rhododendron sp.: Discosia rhododendricola sp.nov, Neopestalotiopsis rhododendricola sp.nov and Diaporthe nobilis as a New Host Record."

_jof, 2022, doi:10.3390/jof8090907_

Round 1

Reviewer 1 Report

Dear Author

#The comments are highlighted in the PDF file.

-The two new species Discosia rhododendronicola and Neopestalotiopsis rhododendricola do not have high support value in the phylogenbetic tree. I suggest the authors add the sequence data of more isolates for these two new taxa to the phylogeny and if they made distinct clade with good support value, then introduce them as new species.

-Please also add the ex-type number (living culture) of the new taxa and also add the Mycobank numbers.

- Please take care about the italic format. All the taxonomic ranks should be in italic. Gene names should be in italic.

With kind regards

Reviewer 2 Report

The lines numbers are missing. Therefore, I will transcribe the totality of the text.

Abstract: In the present study, we report two new asexual fungal species (i.e., Discosia rhododendronicola, Neopestalotiopsis rhododendricola; Sporocadaceae) and a new host for a previously described species (i.e., Diaporthe nobilis; Diaporthaceae). All species were isolated from Rhododendron spp. in Kunming (Yunnan Province, China). All taxa are described based on morphology, and phylogenetic relationships were inferred using a multigenic approach (LSU, ITS, RPB2, TEF1, and TUB2). The phylogenetic analyses indicated that D. rhododendronicola sp. nov is phylogenetically related to D. muscicola and N. rhododendricola sp. nov to N. sonnaratae. Diaporthe nobilis reported herein as a new host record for China, and its phylogeny is depicted based on ITS, TEF1, and TUB2 sequence data.

The first sentence gave a 6% plagiarism.

Please consider rephrasing the last sentence, it is rather confusing. Moreover, please indicate the new host.

1. Introduction

Rhododendron, a genus of shrub and small to large trees belonging to Ericaceae, are indicators of health for forest areas [1], commonly found in low-quality acidic soil and sterile conditions. The plant is mainly distributed in India and southeastern Asia, extending from the northwest Himalayas (Arunachal Pradesh to Bhutan, eastern Tibet, Nepal, north Myanmar, Sikkim, and west central China) [4]. Rhododendron flowers are used as food, to produce fermented wine, and to make herbal tea due to their distinctive flavor and color [2, 3]. Fungi colonizing Rhododendron include Alternaria alternata, Aspergillus brasiliensis, Chrysomyxa dietelii, C. succinea [5], Diaporthe nobilis [6], Epicoccum nigrum, Mucor hiemalis, Pestalotiopsis sydowiana, and Trichoderma koningii [7]. However, given the economic importance of this plant, it is imperative to assess the fungal species associated with it.

Discosia, introduced in Discosiaceae by Maharachchikumbura et al. [9], accommodates the type genus Discosia and the type species D. artocreas. Senanayake et al. [10, 11] introduced Adisciso, Discosia, Discostroma, Immersidiscosia, Sarcostroma, and Seimatosporium in the Discosiaceae family. Jaklitsch et al. [12] considered Discosiaceae a synonym of Sporocadaceae based on DNA sequence analyses with strong phylogenetic support. Wijayawardene et al. [13] accepted Discosia species belonging to the family Sporocadaceae. Libert introduced Discosia, in 1837, being Discosia strobilina the lectotype [10, 13]. Liu et al. [14] reviewed the generic description of Discosia, an updated morphology, and the phylogenetic relationships based on ITS sequence data [14]. There are 118 epithets of Discosia in Index Fungorum 2022 [15]. Discosia has been identified as an asexual fungus characterized by uni- to multilocular conidiomata with multi-layered walls. Conidiogenous cells are monoblastic, phialidic to annellidic. Conidial types are bipolar, polar, and subpolar appendages, usually hyaline to pale brown [10].

The genus Neopestalotiopsis, introduced by Maharachchikumbura et al. (2014) [16], belongs to the family Sporocadaceae (Amphisphaeriales,  Sordariomycetes) [8, 9, 16], being N. protearum the type species. Neopestalotiopsis species have been reported on saprobes, trees, or plant pathogens that cause postharvest diseases (e.g., fruit rots and leaf blights) [17, 18]. The sexual morph of Neopestalotiopsis species remains unknown [16, 18, 19, 20]. Neopestalotiopsis species present a worldwide distribution [16, 17, 18, 19, 20, 21, 22]. The genus has also been reported in caves in China [ref.]. Studies related to the taxonomy of Neopestalotiopsis included DNA sequence analyses and phylogeny using ITS, TEF1, and TUB2 [22].

The genus Diaporthe, introduced by Nitschke [23], belongs to the family Diaporthaceae (Diaporthales, Sordariomycetes) [9, 10]. Diaporthe species present a worldwide distribution [6, 24, 25, 26, 27, 28]. The genus has been associated with several grapevine diseases in Europe [29] and was detected in Uruguayan deciduous fruit tree (Malus domestica 'Gala') wood diseases [30]. Studies related to the taxonomy of Diaporthe included DNA sequence analyses and phylogeny of the ITS, TEF1, TUB2, and CAL loci [6, 31]. Dissanayake et al. [32] provided phylogenetic relationships of 171 Diaporthe species currently known from culture or direct sequencing and are linked to their holotype, epitype, isotype, or neotype, and that can now be recognized with DNA sequence data, essential to species identification [33].

In this study, we introduce new species D. rhododendronicola, N. rhododendricola, and a new host record of D. nobilis, collected from dead leaves of Rhododendron species in China. We further provide descriptions, illustrations, and DNA sequence-based phylogeny to verify identification and placement.

Rhododendron species present a wider distribution. For example, these plants can be found native to several European regions.

Be consistent in the description of the species included.

Moreover, the text is quite convoluted. Please consider simplifying.

Avoid using more than 3 bibliographic references per sentence (prefer revision articles to avoid the extensive usage of bibliographic references)

 2. Materials and Methods

2.1. Sample Collection, Morphological Observation, and Fungal Isolation

Isolation was performed as described by Senanayake et al. [34]. Dead leaves of Rhododendron spp. were collected from Kunming (Yunnan Province, China) and brought to the laboratory in labeled paper envelopes. A light microscope (Nikon ECLIPSE 80i compound microscope) was used to observe the specimens. Spore mass fruiting bodies were isolated on potato dextrose agar (PDA) plates and incubated at 25 °C.

The isolates were transferred to new PDA plates, incubated at 25 °C, and photographed using a Canon EOS 600D digital camera fitted to the microscope. The Tarosoft (R) Image Frame Work program measured the morphological characteristics. The figures were processed using Adobe Photoshop CS6 Extended version 10.0 (Adobe Systems, USA).

The specimens were deposited at the Herbarium of Mae Fah Luang University (Herb. MFLU), and living cultures at the Culture Collection of Mae Fah Luang University (MFLUCC) Chiang Rai, Thailand. Facesoffungi and Index Fungorum data are also provided [35, 36]. New species were established based on guidelines provided by Jeewon and Hyde [37].

2.2. DNA Extraction, PCR Amplification, and Sequencing

Fungal cultures were grown on PDA, at 25 °C, for 2–4 weeks. The Biospin Fungus Genomic DNA Extraction Kit-BioFlux (BioFlux®, China) was used to extract DNA from the mycelium. PCR amplification was performed using the primer pairs ITS4/ITS5 for the internal transcribed spacer region of ribosomal DNA [38], LR0R/LR5 for large subunit nuclear ribosomal DNA [39], EF-728F/EF-986R for translation elongation factor 1-alpha gene [40], fRPB2-5f/fRPB2-7cR for the second largest subunit of RNA polymerase [41], and Bt2a/Bt2b for beta-tubulin [42]. The PCR conditions were based on the methodology described by Chaiwan et al. [43].

2.3. Phylogenetic Analyses

The sequence alignment and phylogenetic analyses were performed as outlined by Dissanayake et al. [44] and Chaiwan et al. [43, 45, 46]. Phylogenetic analyses were performed using a combined Discosia dataset of ITS, LSU, RPB2, TEF1, and TUB2 sequence data combined with Neopestalotiopsis and Diaporthe dataset of ITS, TEF1, and TUB2 sequence data. Taxa used in the analyses were obtained through recent publications [8, 28, 60]. The phylogenetic analyses were carried out using maximum parsimony (MP), maximum likelihood (ML), and Bayesian posterior probabilities (BYPP). PAUP v4.0b10 was used to conduct the parsimony analysis to obtain the phylogenetic trees [47]. Trees were inferred using the heuristic search option with 1000 random sequence additions. Maxtrees were set to 1000, branches of zero length were collapsed, and all multiple parsimonious trees were saved. Descriptive tree statistics for parsimony; Tree Length [TL], Consistency Index [CI], Retention Index [RI], Relative Consistency Index [RC], and Homoplasy Index [HI] were calculated for trees generated following the Kishino-Hasegawa test (KHT) criteria [48], which was performed to determine whether trees were significantly different. Maximum-parsimony bootstrap values equal to or greater than 60% are given as the second set of numbers above the nodes (Fig. 1). Maximum likelihood analysis was performed by using RAxML-HPC2 on XSEDE (8.2.8) [46, 48, 49, 50]. The search strategy was set to rapid bootstrapping, and the analysis was carried out using the GTRGAMMAI model of nucleotide substitution. Maximum Likelihood bootstrap values equal to or greater than 60% are given as the first set of numbers above the nodes (Fig. 2, 3). Bayesian Inference (BI) analysis was conducted with MrBayes v. 3.1.2 to evaluate the posterior probabilities (BYPP) by Markov Chain Monte Carlo sampling [51]. Two parallel runs were conducted using the default settings but with the following adjustments: Six simultaneous Markov chains were run for 2,000,000 generations, and trees were sampled every 200 generations. The distribution of log-likelihood scores were examined to determine the stationary phase for each search and to decide if extra runs were required to achieve convergence, using the program Tracer 1.4 [52]. The first 10% of generated trees were discarded, and the remaining 90% of trees were used to calculate the majority rule consensus tree's posterior probabilities (PP). The phylogenetic trees were viewed in FigTree v. 1.4 [53] and edited using Microsoft Office PowerPoint 2007 and Adobe Photoshop CS6 Extended [43]. The finalized alignment and tree obtained in this study were deposited in TreeBASE (www.treebase.org).

2.4. Phylogenetic Species analysis

The related species were analyzed using the Genealogical Concordance Phylogenetic Species Recognition model. A pairwise homoplasy index (PHI) [105] is a model test based on the fact that multiple gene phylogenies will be concordant between species and discordant due to recombination and mutations within a species. The data were analyzed by the pairwise homoplasy index (PHI) test [105]. The test was performed in SplitsTree4 [106, 107] as described by Quaedvlieg [108] to determine the recombination level within phylogenetically closely related species using a five-locus concatenated dataset to determine the recombination level within phylogenetically closely related species. If the PHI is below a 0.05 threshold (Фw<0.05), it indicates significant recombination in the dataset. This means that related species in a group and recombination levels are not different. If the PHI is below a 0.05 threshold (Фw>0.05), it indicates that it is not significant, which means the related species in a group level are different. The new species and its closely related species were analyzed using this model. The relationships between closely related species were visualized by constructing a split graph, using both the LogDet transformation and splits decomposition options (Fig. 5).

2.5. Discosia, habitat, and known distribution checklist associated with Rhododendron sp.

An updated checklist of Discosia based on the SMML database (https://nt.arsgrin.gov/fungaldata bases/) (latest accessed 10-6-2022) is provided [92]. Those species for which molecular data is available are indicated. The distribution information from type or original descriptions available and the locality from where Discosia have been recorded on Rhododendron spp. is provided, including all those encountered during this study.

Please consider removing the reference to Figures in the Material and Methods section.

If the Methods were used exactly as described by other authors is enough to refer to the article (no need to describe them)

Avoid using more than 3 bibliographic references per sentence (prefer revision articles to avoid the extensive usage of bibliographic references).

The bibliographic references are not continuously presented.

Some Plagiarism was detected.

3. Results

3.1. Phylogenetic analyses

The combined sequence alignments of Discosia comprised 53 taxa (Table 1), with Immersidiscosia eucalypti MFLU16–1372 and NBRC 104195 as the outgroup taxa. The dataset comprised 4671 characters, including alignment gaps (LSU, ITS, RPB2, TEF1, and TUB2 sequence data) after alignment, of which 879 characters were derived from LSU, 544 characters from ITS, 1061 characters from RPB2, 614 characters from TEF1, and 1573 characters from TUB2. The MP analysis for the combined dataset had 413 parsimony informative, 3662 constant, 596 parsimony uninformative characters and yielded a single most parsimonious tree (TL = 1573, CI = 0.793, RI = 0.735, RC=0.583; HI = 0.207). The RAxML analysis of the combined dataset yielded the best scoring tree with a final ML optimization likelihood value of – 13832.959754. The matrix had 945 distinct alignment patterns, with 68.54% of undetermined characters or gaps. Estimated base frequencies were as follows: A = 0.247134, C = 0.242520, G = 0.244931, T = 0.265415; substitution rates AC = 1.869195, AG = 4.658274, AT = 1.756217, CG = 1.367895, CT = 8.434170, GT = 1.000000; gamma distribution shape parameter α = 0.196640. Bayesian posterior probabilities from Bayesian inference analysis were assessed with a standard deviation of split frequencies = 0.009983. The phylogenetic tree in this study showed that D. rhododendronicola KUN-HKAS 123205 is phylogenetically related to D. muscicola, with low bootstrap support (Fig. 1). Sequence alignments were deposited in TreeBASE (TreeBASE ID: http://purl.org/phylo/treebase/phylows/study/TB2:S27978?x-accesscode=2cf8d87414a70823533d7b67a4cccfe7&format=html).

The combined sequence alignments of Neopestalotiopsis comprised 88 taxa (Table 2), with Monochaetia monochaeta CBS115004 and M. ilexae CBS101009 used as the outgroup taxa. The dataset comprised 2963 characters, including alignment gaps (ITS, TUB2, and TEF1 sequence data). After the alignment, 773 characters were derived from ITS, 1004 from TUB2, and 1186 from TEF1. The MP analysis for the combined dataset had 629 parsimony informative, 1776 constant, 558 parsimony uninformative characters and yielded a single most parsimonious tree (TL = 2381, CI = 0.690, RI = 0.812, RC=0.561; HI = 0.310). The RAxML analysis of the combined dataset yielded the best scoring tree with a final ML optimization likelihood value of – 16300.542588. The matrix had 1352 distinct alignment patterns, with 42.54% of undetermined characters or gaps. Estimated base frequencies were as follows: A = 0.236392, C = 0.276653, G = 0.218159, T = 0.268796; substitution rates AC = 1.159931, AG = 2.997984, AT = 1.233611, CG = 0.970177, CT = 3.579206, GT = 1.000000; gamma distribution shape parameter α = 0.494624. Bayesian posterior probabilities from Bayesian inference analysis were assessed with a standard deviation of split frequencies = 0.024223. The phylogenetic tree in this study showed that N. rhododendricola KUN-HKAS 123204 belonged to a separate clade, phylogenetically related to N. sonneratae, N. coffeae-arabicae, and N. thailandica with 88% MP support (Fig. 2). Sequence alignments are deposited in TreeBASE (TreeBASE ID: http://purl.org/phylo/treebase/phylows/study/TB2:S28808?x-accesscode=db10f746c791f07237bf97ce69375129&format=html).

The combined sequence alignments of Diaporthe comprised 56 taxa (Table3), with Diaporthella corylina CBS 121124 used as the outgroup taxon. The dataset comprised 2350 characters, including alignment gaps (ITS, TEF1, and TUB2 sequence data). After alignment, 641 characters were derived from ITS, 916 from TEF1, and 793 from TUB2. The MP analysis for the combined dataset had 730 parsimony informative, 1216 constant, 404 parsimony uninformative characters and yielded a single most parsimonious tree (TL = 3968, CI = 0.480, RI = 0.622, RC=0.298; HI = 0.520). The RAxML analysis of the combined dataset yielded the best scoring tree with a final ML optimization likelihood value of – 21299.667012. The matrix had 1319 distinct alignment patterns, with 37.51% of undetermined characters or gaps. Estimated base frequencies were as follows: A = 0.226983, C = 0.316389, G = 0.231894, T = 0.224734; substitution rates AC = 1.153998, AG = 3.111864, AT = 1.039115, CG = 0.869376, CT = 4.271324, GT = 1.000000; gamma distribution shape parameter α = 0.376625. Bayesian posterior probabilities from Bayesian inference analysis were assessed with a standard deviation of split frequencies = 0.009867. The phylogenetic tree in this study showed that D. nobilis KUN-HKAS 123203 grouped with the ex-type strain of D. nobilis, and formed a supported clade with 0.99 PP (Fig. 3). Sequence alignments were deposited in TreeBASE (TreeBASE ID: http://purl.org/phylo/treebase/phylows/study/TB2:S28807?x-accesscode=2e63e307ca3dac808492ae6bd857bb74&format=html).

3.2. Taxonomy

3.2.1. Discosia rhododendronicola Chaiwan & K.D. Hyde, sp. nov. (Figure 4)

Index Fungorum number: 558124

Facesoffungi number: FoF 09452

Etymology: name reflects the host from which the fungus was isolated. Holotype: KUN-HKAS 123205 Saprobic on dead leaves of Rhododendron sp.

Sexual morph: undetermined.

Asexual morph: Conidiomata 200–250 × 30–75 μm, pycnidial, cervular, applanate to disc-like, partly immersed or superficial, black, rounded to irregular in outline, glabrous, unilocular or divided into several locules by tissue conspicuous at the surface. Conidiophores were observed arising from the base, hyaline, filiform to cylindrical, smooth, and reduced to conidiogenous cells. Conidiogenous cells appeared subcylindrical, flask-shaped, hyaline, smooth, phialidic, each producing a single conidium unbranched. Conidia 20–30 × 4–5 μm (xÌ…= 25 × 4.5 μm, n=30), subcylindrical, slightly curved, 3-septate, with slight constrictions at the septa, brown, smooth-walled with unequal cell; bipolar appendages; with a long, tubular base, two median cells subcylindrical, second cell joined to the base 10–15 μm (xÌ…= 12.5 μm) long, the third cell joined to the apex 11–15 μm (xÌ…= 13 μm) long; apical cell subconical with a rounded apex; apical and basal cells each with a subapical, unbranched, filiform, straight appendage; apical appendage 9–11 μm (xÌ…= 10 μm), basal appendage 20–25 μm (xÌ… = 22.5 μm). Culture characteristics: Colonies grown on PDA were filamentous, raised, filiform margin, reached 4–5 cm in 5 days at 25 °C, brown to black, mycelium superficial, branched, septate, white mycelium with aerial on the surface, produced black spore mass.

Material examined: CHINA, Kunming Yunnan Province; on dead leaves of Rhododendron sp. (Ericaceae), 28 July 2018, Napalai Chaiwan, KIB009 (KUN-HKAS 123205, holotype).

Notes: D. rhododendronicola was characterized as similar to D. macrozamiae CPC 32109 [91] concerning conidiomata size (D. rhododendronicola: 200–250 μm diam., 30–75 μm height; D. macrozamiae CPC 32109: 250 μm diam, xx–xx μm height). The species D. rhododendronicola and D. artocreas (type species) shared similar characteristics of conidiophores lining the inner cavity (0–2-septate, rarely branched at base). There were also similar conidial characteristics (conidia between 30–32 μm; the second cell joining to the base was 10–15 μm (xÌ…= 12.5 μm) in D. rhododendronicola; 10–11 μm (xÌ…= 10.5 μm) in D. macrozamiae CPC 32109; the third cell joining to the apex was 11–15 μm (xÌ…= 13 μm) in D. rhododendria, and 4–5 μm (xÌ…= 4.5 μm) in D. macrozamiae CPC 32109. The apical appendage of D. rhododendronicola was 9–11 μm (xÌ…= 10 μm), while in D. macrozamiae (CPC 32109) was 7–11 μm (xÌ…= 9 μm). The basal appendage in D. rhododendronicola was measured to be 20–25 μm (xÌ…= 22.5 μm) in length, and in D. macrozamiae (CPC 32109) 10– 16 μm (xÌ…= 13 μm). Discosia rhododendronicola differed from the type species in ascomata size (D. rhododendronicola: 200–250 μm diam., 30–75 μm height; D. artocreas 150–500 μm diam, 60 μm height). The two species shared similar characteristics of conidiophores and conidiogenous cells. However, D. rhododendronicola had hyaline to pale brown conidiogenous cells and conidia, whereas D. artocreas had hyaline conidiogenous cells and conidia. The second cell joining to the base measured 10–15 μm (xÌ…= 12.5 μm) in D. rhododendronicola, 5–9 μm (xÌ…= 7.5 μm) in D. artocreas. The third cell joining to the apex was 11–15 μm (xÌ…= 13 μm) in D. rhododendria and 3–6 μm (xÌ…= 4.5 μm) in D. artocreas. The apical appendage of D. rhododendronicola was 9– 11 μm (xÌ…= 10 μm), while in D. artocreas was 6–12 μm (xÌ…= 10 μm). The basal appendage in D. rhododendronicola was 20–25 μm (xÌ…= 22.5 μm), while in D. artocreas was 7–12 μm (xÌ…= 10 μm).

The NCBI BLAST search of the ITS sequence from D. rhododendronicola presented a 95.32% similarity with I. eucalypti. A comparison of the 542 ITS (+5.8S) nucleotides of D. rhododendronicola sp. nov. differed from I. eucalypti in 21 (3.87%) nucleotides. We compared 876 LSU nucleotides of D. rhododendronicola with D. muscicola CBS 109.48 and a 0.34% bp difference was observed (a difference of 3 bp in a total 879 bp) (Table 4). When analyzing the gene sequenced, D. rhododendronicola sp. nov. was phylogenetically related to D. macrozamiae CPC 32109, D. muscicola CBS 109.48, D. pleurochaeta KT2179, D. pleurochae KT 2188 and KT 2192, D. tricellularis MAFF237478 and NCBR32705, and D. yakushimensis MAFF 242774 were found to be in a clade with low bootstrap support (Fig 1). The ITS and LSU base pair differences between D. rhododendronicola and other related species are shown in Table 3.

In addition, the PHI test showed Фw = 1.00. This analysis determines the recombination level within phylogenetically closely related species. If the PHI is below a 0.05 threshold (Фw > 0.05), it indicates no significance; the combined phylogenetic analyses provided evidence that the isolate belongs to a new species. Discosia rhododendronicola KUN-HKAS 123205 is phylogenetically related to the clade consisting of D. muscicola CBS 109.48, D. tricellularis MAFF 237478, NBRC 32705, and D. yakushimensis MAFF 242774 (Fig. 5). The results of molecular analyses based on the Genealogical Concordance Phylogenetic Species Recognition (GCPSR) tree support the congruency between of single gene tree and the combined phylogenetic tree. Our GCPSR analyses also showed that D. rhododendronicola MFLU 20–KUN-HKAS 123205 was distinguished as a separate species by genealogical concordance (PHI=1.0). The result showed the same direction as the graph of combined gene similarity of a phylogenetic tree.

3.2.2. Neopestalotiopsis rhododendricola Chaiwan & K.D. Hyde, sp. nov. (Figure 6)

Index Fungorum number: IF558797

Facesoffungi number: FoF 1047

Etymology: name reflects the host from which the fungus was isolated.

Holotype: KUN-HKAS 123204 Saprobic on dead leaves of Rhododendron sp.

Sexual morph: Undetermined.

Asexual morph: Conidiomata (on PDA) 60–80 × 50–75 μm, pycnidial, cervular, applanate to disclike, partly immersed or superficial, globose to clavate, solitary or confluent, embedded or semi-immersed to erumpent, dark brown, exuding globose, dark brown to black conidial masses, rounded to irregular in outline, glabrous, unilocular or divided into several locules by tissue cells. Conidiophores are indistinct, arising from the base, hyaline, filiform to cylindrical, smooth, and often reduced to conidiogenous cells. Conidiogenous cells appeared subcylindrical, flask-shaped, hyaline, smooth, and phialidic, each producing a single conidium. Conidia 20–30 × 5–7 μm (xÌ…= 25 × 6 μm, n=30), subcylindrical fusoid, ellipsoid, straight to slightly curved, 4-septate, (19–28) × 5–7 μm (xÌ…= 23.5 × 6 μm, n=30), μm; basal cell conic with a truncate base, hyaline, rugose and thin-walled, with constrictions at the septa, hyaline, smooth-walled; with a long, tubular base, two median cells subcylindrical, second cell joined to the base 10–15 μm (xÌ…= 12.5 μm) long, the third cell joined to the apex 11–15 μm (xÌ…= 13 μm) long; apical cell subconical with a rounded apex; apical and basal cells each with a subapical, unbranched, filiform, straight appendage; apical appendage 9–11 μm (xÌ… = 10 μm), basal appendage 20–25 μm (xÌ…= 22.5 μm).

Culture characteristics: Colonies grown on PDA, with an undulate edge, reached 4–5 cm in 5 days at 25 °C, mycelium superficial, branched, septate, white mycelium with aerial on the surface, produced black spore mass.

Material examined: CHINA, Kunming Yunnan Province; on dead leaves of Rhododendron sp. (Ericaceae), 28 July 2018, Napalai Chaiwan, KIB008 (KUN-HKAS 123204, holotype).

Notes: Neopestalotiopsis rhododendricola was isolated from a Rhododendron spp. in China. In the phylogenetic analyses, N. rhododendronicola from a distinct lineage sister to N. sonneratae (MFLUCC17-1745T, MFLUCC17-1744), N. coffeae-arabicae (HGUP4019T, HGUP4015) and N. thailandica (MFLUCC17-1730T, MFLUCC17-1731) (Fig. 2). N. sonneratae was reported on leaf spots on Sonneronata alba in Thailand [21], N. thailandica was reported on leaf spots of Rhizophora mucronata Lam. in Thailand [21], and N. coffeae-arabicae was found on leaves of Coffea arabica in China [72]. Neopestalotiopsis rhododendricola sp. nov. resembles N. thailandica in having a similar conidial size [21], but the difference is that N. rhododendricola has 2–3 tubular appendages on the apical cell while N. thailandica showed only 1–2 tubular appendages on the apical cell. Comparison of ITS sequence differences 2 base pair, TEF sequence differences 15 base pair and TUB differences 6 base pair of N. rhododendricola and N. thailandica shows. Therefore, based on morphology and phylogeny, we justify the description of N. rhododendricola as a new species in the Neopestalotiopsis genus.

3.2.3. Diaporthe nobilis Tanaka & S. Endô, in Endô, J. Pl. Prot. Japan 13: (1927) (Figure 7)

Index Fungorum number: IF 265419

Facesoffungi number: FoF 02717 Saprobic on dead leaves of Rhododendron sp.

Sexual morph: Undetermined.

Asexual morph: Conidiomata pycnidial 50–100 × 25–75 μm. (xÌ…= 75 × 50 μm, n=10), globose to stromatic, multilocular, dark brown to black, scattered. Conidiophores were observed arising from the base, hyaline, filiform to cylindrical, smooth, and straight. Conidiogenous cells 35–40 × 1–2 μm (xÌ…= 37.5 × 1.5 μm, n=10), phialidic, cylindrical, terminal, and lateral, slightly tapered towards the apex, with visible periclinal, thickening, hyaline, smooth-walled. Beta conidia 16–20 × 1–2 μm (xÌ…= 18 × 1.5 μm, n=30), hyaline smooth, guttulate, fusoid to ellipsoid, straight, tapered towards both ends, apex sub obtuse, base sub truncate, aseptate. Alpha conidia not found. Culture characteristics: Colonies grew on PDA, filamentous, flattened, dense, and felty, reaching 5–6 cm in 14 days at 25 °C, white to brown on the surface, mycelium superficial, branched, septate.

Material examined: CHINA, Kunming Yunnan Province, on dead leaves of Rhododendron sp. (Ericaceae), 28 July 2018, Napalai Chaiwan, KIB003 (KUN-HKAS 123203, new host record) living culture MFLUCC 18–1482.

Notes: Diaporthe nobilis MFLUCC 18–1482 clustered with D. nobilis CBS 587.79 and CBS113470 with high 0.99 PP bootstrap support. Conidiomata from the strain MFLUCC 18–1482 was acervular, semi-immersed, globose to eustromatic, and multilocular, while the other strain was pycnidia subcuticular, scattered to confluent, and uniloculate. Our strain was observed to share similar morphological characteristics with other D. nobilis strains in having conidiogenous cells formed at the apex of the conidiophores, cylindric, straight, or curved hyaline and smooth-walled. Comparison of ITS, TEF1 and TUB2 sequence data of the isolate MFLUCC 18–1482 and D. nobilis CBS113470, revealed 9 bp (1.41%) in 637 ITS (+5.8S) nucleotides, 2 bp (0.40%) in 496 TEF1 nucleotide and 6 bp (0.71%) in 844 TUB2 nucleotide. Therefore, we consider our strain (MFLUCC 18–1482) as D. nobilis and a new host record from Rhododendron sp. in China.

4. Discussion

Discosia species are distributed on various vascular plants and a wide range of hosts and occur primarily in their asexual state as endophytes, saprobes, and pathogens [20, 92]. Host-specificity of the taxa in this genus has not yet been established. Discosia species can be found on Fagus sylvatica (Fagaceae), Gaultheria procumbens (Ericaceae), Platanus orientalis (Platanaceae), Quercus sp. (Fagaceae), Syzygium cumini (Myrtaceae), Smilax rotundifolia (Smilacaceae), and leaves of undetermined plants [55]. Discosia blumencronii Bubák was reported from Rhododendron poniicum [89], while other species can be found on leaves of Beilschmeidia tarairi (Lauraceae), Brachychiton populneus (Malvaceae), Ceanothus fiedleri (Rhamnaceae), Eucalyptus sp. (Myrtaceae), Laurus nobilis (Lauraceae), and Phillyrea latifolia (Oleaceae) [10, 55]. Discosia species is distributed in temperate regions, being previously reported in Algeria, Austria, Brazil, France, Germany, India, Italy, New Zealand, Portugal, USA, Sweden, Tunisia, and Turkey [10]. The new taxon, D. rhododendronicola, was phylogenetically related to D. muscicola, described by NicotToulouse Morelet (1968), and isolated from Cephalozia bicuspidate (Cephaloziaceae) in France. Nonetheless, no morphological data is available for comparison [91]. Discosia rhododendronicola sp. nov. was isolated from Rhododendron sp, and morphology was compared. The ascomata and conidia of D. rhododendronicola were larger than those of D. artocreas, whereas the size of conidiophores, conidiogenous cells, and apical appendage was similar. Discosia rhododendronicola was similar to D. macrozamiae CPC 32109 [57], but the phylogenetic tree showed that our species was more closely related to D. muscicola CBS 109.48. However, from D. muscicola CBS 109.48 only DNA sequence data were available (Fig. 1). It should be pointed out that when the ITS DNA sequences of D. muscicola were subjected to a blast search, the closest hits were Aspergillus species similar to A. avenaceus. Our novel species have DNA sequence data from 3 regions (LSU, ITS, and RPB2), but we can only compare the LSU region for D. muscicola CBS 109.48 as there is no sequence data of the protein-coding gene available for comparison. Based on the previous study of Wijayawardene et al. [8], 34 genera are recognized in Sporocadaceae. In this study, we introduce D. rhododendronicola as a new species based on the phylogenetic and pairwise homoplasy index. Discosia species share similar morphological characters, but most characters are not meaningful in species delineation. In this study, our new species constitutes a different branching pattern in our phylogeny and appears distinct from extant species. A relationship among species based on similar conidial characters does not necessarily correlate with our phylogenetic relationships, and this indicates that morphology has little significance for reliable species identification. Further studies are also needed to determine whether Discosia species might be host-specific. Herein, we introduce a new species N. rhododendricola KUN-HKAS 123204, within the Neopestalotiopsis genus that was separated from the other Neopestalotiopsis clade and based on morphological and molecular phylogenetic analyses (Fig. 2). Neopestalotiopsis are characterized by their conidia with versicolor median cells, by indistinct conidiogenous cells [16] and the ITS, TUB2 and TEF1 sequences. The newly described species is phylogenetically related to the group of N. sonneratae, N. coffeae-arabicae, and N. thailandica in the phylogenetic tree (Fig. 2), and the relationship is not strongly supported, likely because of the source of the habitat of each species. Our new species was found on Rhododendron sp. plant host from China, while N. sonneratae was reported on leaf spots on Sonneronata alba L. [21], N. thailandica was reported on leaf spots of Rhizophora mucronata Lam. both strains have been reported in Thailand [21], and N. coffeae-arabicae were found on leaves of Coffea arabica in China [72]. Diaporthe species have been reported as plant pathogens, saprobes, and endophytes on many plant hosts [23, 28, 92]. Species of Diaporthe are not host-specific [6, 28, 41]. Substrates colonized by members of Diaporthe recorded until now are mainly in dicotyledons of Ericaceae, Fagaceae, Pinaceae, Rhizophoraceae, Rosaceae, and Theaceae. Some species of Diaporthe can be found on more than one host. For example, D. nobilis was reported on Camellia sinensis (Theaceae), Castanea sativa (Fagaceae), Malus pumila (Rosaceae), Pinus pantepella (Pinaceae), Pyrus pyrifolia (Rosaceae) and Rhododendron sp. (Ericaceae) [6, 28, 41, 92].

Diaporthe is mostly presented in the asexual morph as coelomycetes [23]. Diaporthe nobilis complex [6] has alpha and beta conidia [28]. However, our strain was only found to have beta conidia. Diaporthe has been reported on Rhododendron spp. from Europe (Latvia) [6]. The strain MFLUCC 18–1482 was found isolated from Asia (China), indicating that the species is distributed in different geographical locations on the host; however, there is a need for more collections of microfungi associated with Rhododendron, targeting a wide variety of geographical locations. A checklist for Discosia species associated with Rhododendron is also provided herein.

4. The checklist of Discosia, habitat, and known distribution associated with Rhododendron sp.

Based on the USDA Systematic Mycology and Microbiology Laboratory (SMML) database [92], with relevant literature and the author’s studies. The current name and fungal classification were used according to Index Fungorum (2022) [15], an outline of Ascomycota [8].

*Species confirmed with molecular data are marked with an asterisk.

1. Discosia artocreas (Tode) Fr., Summa veg. Scand., Sectio Post. (Stockholm): 423 (1849)

= Sphaeria artocreas Tode, F. Meckl. 2: 77, 1791; Fries, Syst. Myc. 2: 523, 1823.

Habitat: Rhododendron arboretum, R. campylocarpum, R. nudiflorum [93, 94], R. catawbiense, R. maximum [95], R. ponticum [96, 97] and Rhododendron sp. [93, 94, 98]

Known distribution: Italy [96], Maryland [93, 94, 95], New York [95], United Kingdom [98], Turkey [97*], Washington [93, 94].

2. Discosia blumencronii Bubák, in Handel-Mazzetti, Annln K. K. naturh. Hofmus. Wien 23: 106 (1910)

Habitat: Rhododendron ponticum (on dead leaves) [99]

Known distribution: Turkey [99]

3. Discosia himalayensis Died., Annls mycol. 14(3/4): 218 (1916)

= Discosia strobilina Lib. ex Sacc., Syll. Fung. (Abellini) 3: 656 (1884)

Habitat: Rhododendron arboretum, R. campanulatum (on dead leaves) [99, 100, 101]

Known distribution: India [99, 100, 101]

4. Discosia rhododendri (Speschnew, Monit. Jard. Bot. Tiflis 4: 10 (1906)

Habitat: Rhododendron albrechtii (on dead leaves) [102*], R. ponticum [103*], Rhododendron sp. (on leaves) [99]

Known distribution: Japan [102*], Turkey [103*]

5. Discosia rhododendronicola (This study*)

Habitat: Rhododendron sp. (on dead leaves) (This study*)

Known distribution: China (This study*)

6. Discosia sp.

Habitat: Rhododendron sp. [102*]

Known distribution: Japan [102*]

7. Discosia tricellularis (Okane, Nakagiri & Tad. Ito) F. Liu, L. Cai & Crous, in Liu, Bonthond, Groenewald, Cai & Crous, Stud. Mycol. 92: 322 (2018) [2019] J. Fungi 2021, 7, x FOR PEER REVIEW 11 of 35

Habitat: Rhododendron indicum [104]

Known distribution: Japan [104]

8. Discosia vagans De Not., Atti Acad. Tor.: 354 (1849)

Habitat: Rhododendron arboretum, R. nilagiricum, R. veitchianum [54*, 101], R. ponticum [56*]

Known distribution: India [103, 104*], Scotland [54*]

Round 2

Reviewer 1 Report

Dear author

The paper is prepared well.

All the best

Author Response

I improved following your comments.
